# Structural basis of EHEP-mediated offense against phlorotannin-induced defense from brown algae to protect *aku*BGL activity

Xiaomei Sun[1†], Yuxin Ye[1†‡], Naofumi Sakurai[1], Hang Wang[1], Koji Kato[1§], Jian Yu[1], Keizo Yuasa[2#], Akihiko Tsuji[2], Min Yao[1]*

[1]Faculty of Advanced Life Science, Hokkaido University, Sapporo, Japan; [2]Graduate School of Bioscience and Bioindustry, Tokushima University, Tokushima, Japan

*For correspondence:
yao@castor.sci.hokudai.ac.jp

†These authors contributed equally to this work

Present address: ‡Pingshan Translational Medicine Center, Shenzhen Bay Laboratory, Shenzhen, China; §Structural Biology Division, Japan Synchrotron Radiation Research Institute (JASRI), Hyogo, Japan; #Faculty of Science and Engineering, Setsunan University, Neyagawa, Japan

Competing interest: The authors declare that no competing interests exist.

**Abstract** The defensive–offensive associations between algae and herbivores determine marine ecology. Brown algae utilize phlorotannin as their chemical defense against the predator *Aplysia kurodai*, which uses β-glucosidase (*aku*BGL) to digest the laminarin in algae into glucose. Moreover, *A. kurodai* employs *Eisenia* hydrolysis-enhancing protein (EHEP) as an offense to protect *aku*BGL activity from phlorotannin inhibition by precipitating phlorotannin. To underpin the molecular mechanism of this digestive–defensive–offensive system, we determined the structures of the apo and tannic acid (TNA, a phlorotannin analog) bound forms of EHEP, as well as the apo *aku*BGL. EHEP consisted of three peritrophin-A domains arranged in a triangular shape and bound TNA in the center without significant conformational changes. Structural comparison between EHEP and EHEP–TNA led us to find that EHEP can be resolubilized from phlorotannin precipitation at an alkaline pH, which reflects a requirement in the digestive tract. *aku*BGL contained two GH1 domains, only one of which conserved the active site. Combining docking analysis, we propose the mechanisms by which phlorotannin inhibits *aku*BGL by occupying the substrate-binding pocket, and EHEP protects *aku*BGL against this inhibition by binding with phlorotannin to free the *aku*BGL pocket.

## eLife assessment

This **important** study presents **convincing** evidence on how the sea slug *Aplysia kurodai* optimizes its digestion of brown algae, in a classical predator-prey 'arms race' at the molecular level. The experimental protein structures and enzyme assays provide support for the claims of how *A. kurodai* avoids inhibition by algal compounds, and also hold promise for biotechnological applications.

## Introduction

Over millions of years of evolution, predators have successfully coevolved with their prey to maintain an ecological balance (*Becklin, 2008*). In marine habitats, interactions between algae and marine herbivores dominate marine ecosystems. Most algae are consumed by marine herbivores (*Jormalainen and Honkanen, 2008*). They produce secondary metabolites as a chemical defense to protect themselves against predators. In brown algae *Eisenia bicyclis*, laminarin is a major storage carbohydrate, constituting 20–30% of algae dry weight. The sea hare *Aplysia kurodai*, a marine gastropod, preferentially feeds on *E. bicyclis* with its 110 and 210 kDa β-glucosidases (*aku*BGLs), hydrolyzing the laminarin and releasing large amounts of glucose. Interestingly, such a feeding strategy has attracted attention for producing glucose as a renewable biofuel source (*Enquist-Newman et al., 2014*). However, to

protect themselves against predators, brown algae produce phlorotannin, a secondary metabolite, thereby reducing the digestion by *A. kurodai* by inhibiting the hydrolytic activity of *aku*BGLs. As the 110 kDa *aku*BGL is more sensitive to phlorotannin than the 210 kDa BGL (*Tsuji et al., 2017*), we focused on the 110 kDa *aku*BGL in this study (hereafter, *aku*BGL refers to 110 kDa *aku*BGL).

To counteract the antipredator adaptations of algae, herbivores use diverse approaches, such as detoxification, neutralization, defense suppression, and physiological adaptations (*Erb and Reymond, 2019*). *A. kurodai* inhibits the phlorotannin defense of brown algae through *Eisenia* hydrolysis-enhancing protein (EHEP), a protein from their digestive system that protects *aku*BGL activity from phlorotannin inhibition (*Tsuji et al., 2017*). Previous studies have shown that incubating *E. bicyclis* with *aku*BGL in the presence of EHEP results in increased glucose production because EHEP binds to phlorotannin and forms an insoluble complex (*Tsuji et al., 2017*).

The *aku*BGL–phlorotannin/laminarin–EHEP system exemplifies the digestion process of *A. kurodai* as well as the defense and antidefense strategies between *E. bicyclis* and *A. kurodai*. Although the defense/antidefense strategy has been established, the detailed molecular mechanism of this interplay remains unknown. Furthermore, phlorotannin inhibition hinders the potential application of brown algae as feedstocks for enzymatically producing biofuel from laminarin. Thus, understanding the underlying molecular mechanisms will be beneficial for the application of this system in the biofuel industry.

Despite the potential use of laminarin hydrolytic enzymes in the biofuel industry, only a few BGLs of glycoside hydrolases belonging to the GH3 and GH1 family are known to hydrolyze laminarin (e.g., *Talaromyces amestolkiae* BGL (*Méndez-Líter et al., 2018*), *Ustilago esculenta* BGL (*Nakajima et al., 2012*), and *Vibrio campbellii* BGL (*Wang et al., 2015*) from the GH3 family and *Saccharophagus degradans* 2-40$^T$ BGL (*Kim et al., 2018*) from the GH1 family). GH3 is a multidomain enzyme family characterized by N-terminal $(\beta/\alpha)_8$ (NTD) and C-terminal $(\beta/\alpha)_6$ (CTD) domains, with or without auxiliary domains (*Mohsin et al., 2019*); the nucleophile aspartate and the acid/base glutamate residues exist in the NTD and CTD, respectively. In contrast, the members of the GH1 family generally share a single $(\beta/\alpha)_8$-fold domain (hereafter referred to as the GH1 domain [GH1D]), and the two glutamic acid catalytic residues are located in the carboxyl termini of β-strands 4 and 7. Therefore, the two families may use different substrate recognition and catalytic mechanisms for laminarin. Intriguingly, although *aku*BGL possesses laminarin hydrolytic activity and belongs to the GH1 family, its molecular weight is considerably higher than that of other GH1 members. Sequence analysis has indicated that *aku*BGL consists of ≥2 GH1Ds. Because no structural information of BGL active on polysaccharides is available, the catalytic mechanism toward laminarin remains unclear.

There is limited information on EHEP, a novel cysteine-rich protein (8.2% of the amino acid content), because no structurally or functionally homologous protein exists in other organisms. EHEP was predicted to consist of three peritrophin-A domains (PADs) with a cysteine-spacing pattern of $CX_{15}CX_5CX_9CX_{12}CX_{5-9}C$. The PADs consist of peritrophic matrix proteins, which have been proposed to play an important role in detoxifying ingested xenobiotics (*Hegedus et al., 2009*). For instance, *Aedes aegypti* intestinal mucin 1 (*Ae*IMUC1) consists of a signal peptide followed by three PADs with an intervening mucin-like domain; its expression is induced by blood feeding. *Ae*IMUC1-mediated blood detoxification during digestion is completed by binding to toxic heme molecules (*Devenport et al., 2006*). Despite the similar domain organization of EHEP and *Ae*IMUC1, their functions and binding partners are different, implying their different characteristics. However, the characteristics of the EHEP–phlorotannin insoluble complex remain unknown; moreover, it remains unclear why and how EHEP protects *aku*BGL from phlorotannin inhibition.

In this study, we determined the structures of the apo and tannic acid (TNA, phlorotannin analog) bound forms of EHEP (EHEP, EHEP–TNA), as well as *aku*BGL, all isolated from *A. kurodai*. The structure of EHEP consists of three PADs arranged in a triangular shape, with TNA bound at the surface of the triangle center. A structural comparison of EHEP and EHEP–TNA revealed no significant changes in conformation upon TNA binding, implying that EHEP maintains its structure when precipitated with TNA. Then, we found the conditions to resolubilize EHEP–TNA precipitate for EHEP recycling. The obtained *aku*BGL structure suggests that only one GH1D (GH1D2) possesses laminarin hydrolytic activity; subsequently, ligand-docking experiments demonstrated that TNA/phlorotannin has a higher docking score than laminarin. Our results revealed the mechanisms by which EHEP protects *aku*BGL

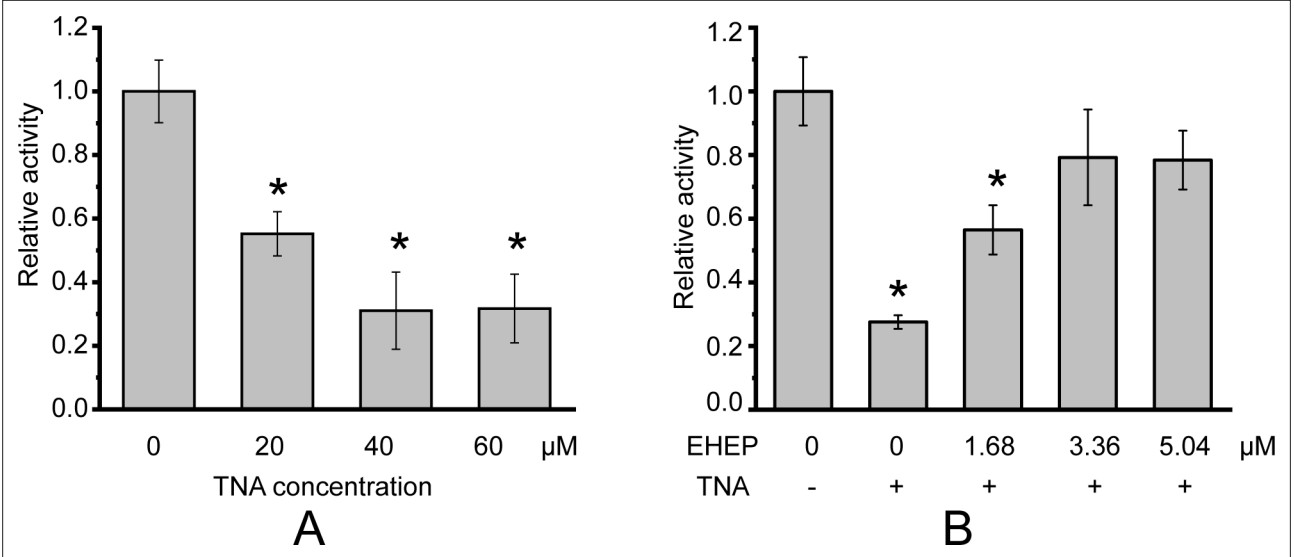

**Figure 1.** Galactoside hydrolytic activity of *aku*BGL toward 2.5 mM ortho-nitrophenyl-β-galactoside. Activity (%) is shown as the fold increase relative to *aku*BGL without the addition of tannic acid (TNA) or *Eisenia* hydrolysis-enhancing protein (EHEP). (**A**) The hydrolytic activity of *aku*BGL (0.049 μM) with TNA at different concentrations. (**B**) The hydrolytic activity of *aku*BGL (0.049 μM) with 40 μM TNA and EHEP at different concentrations. The average and standard deviation of the relative activity were estimated from three independent replicates (*N* = 3). Asterisks in the top of error bars indicated significant (p<0.005) differences by t-test in comparison with 0 μM TNA (A) or 0 μM EHEP and no TNA (B).

The online version of this article includes the following figure supplement(s) for figure 1:

**Figure supplement 1.** High-performance liquid chromatography (HPLC) profiles of *aku*BGL activity toward ortho-nitrophenyl-β-galactoside.

from phlorotannin inhibition and how phlorotannin inhibits the hydrolytic activity of *aku*BGL, providing structural support for the potential application of brown algae in biofuel production.

## Results

### Effects of TNA on *aku*BGL activity with or without EHEP

Phlorotannin, a type of tannin, is a chemical defense metabolite of brown algae. It is difficult to isolate a compound from phlorotannins because they are a group of polyphenolic compounds with different sizes and varying numbers of phloroglucinol units (*Cassani et al., 2020*), such as eckol, dieckol, and so on (*Imbs and Zvyagintseva, 2018*). Previous studies have reported that the phlorotannin-analog TNA has a comparable inhibitory effect on *aku*BGL to phlorotannin (*Tsuji et al., 2017*). Hence, we used TNA instead of phlorotannin to explore phlorotannin binding with EHEP and *aku*BGL. This activity assay system involves multiple equilibration processes: *aku*BGL ⇋ substrate, *aku*BGL ⇋ TNA, and EHEP ⇋ TNA. First, we confirmed TNA inhibition of *aku*BGL activity and clarified the protective effects of EHEP from TNA inhibition. The inhibition experiments showed that the galactoside hydrolytic activity of *aku*BGL decreased with increasing TNA concentration, indicating that TNA inhibits *aku*BGL activity in a dose-dependent manner (*Figure 1A*, *Figure 1—figure supplement 1A*). Approximately 70% *aku*BGL activity was inhibited at a TNA concentration of 40 μM. Moreover, protection ability analysis revealed that EHEP protects *aku*BGL activity from TNA inhibition in a dose-dependent manner, as indicated by the recovery of the inhibited *aku*BGL activity with increasing EHEP concentration (*Figure 1B*, *Figure 1—figure supplement 1B*). Approximately 80% of *aku*BGL activity was recovered at an EHEP concentration of 3.36 μM.

### Overall structure of EHEP

Considering the lack of known homologous proteins of EHEP, we determined the structure of EHEP using the native-SAD method at a resolution of 1.15 Å, with an $R_{work}$ and $R_{free}$ of 0.18 and 0.19, respectively (*Table 1*). The residues A21–V227 in purified EHEP (1–20 aa were cleaved during maturation) were built, whereas two C-terminal residues were disordered. The structure of EHEP consists of three PADs: PAD1 (N24–C79), PAD2 (I92–C146), and PAD3 (F164–C221), which are linked by two long

**Table 1.** X-ray data collection and structure-refinement statistics.

| | EHEP1 | EHEP2 | EHEP–TNA | *aku*BGL |
|---|---|---|---|---|
| Data collection | | | | |
| Beamline | PF BL17A | Spring 8 BL-41XU | PF BL17A | PF BL1A |
| Wavelength (Å) | 0.9800 | 1.0000 | 0.9800 | 1.0000 |
| Resolution range (Å) | 46.56–1.15 (1.20–1.15) | 47.02–1.4 (1.45–1.4) | 46.84–1.9 (1.97–1.9) | 49.65–2.7 (2.80–2.7) |
| Space group | $P2_12_12_1$ | $P2_12_12_1$ | $P2_12_12_1$ | $P6_2$ |
| Unit-cell parameters $a, b, c$ (Å) | 42.2, 65.3, 66.5 | 40.6, 65.6, 67.5 | 42.5, 65.4, 67.2 | 191.7, 191.7, 112.6 |
| Completeness (%) | 93.4 (75.8) | 99.30 (97.98) | 99.9 (99.7) | 99.9 (99.1) |
| Redundancy | 6.6 (6.1) | 6.4 (5.8) | 6.4 (6.4) | 10.7 (10.9) |
| Average $I/\sigma(I)$ | 19.28 (1.97) | 14.34 (3.41) | 15.39 (1.83) | 10.69 (2.57) |
| $R_{meas}$(%)* | 7.3 (90.5) | 8.6 73.7 (55.3) | 8.9 (89.4) | 19.4 (83.0) |
| $CC^{1/2}$ (%) | 99.9 (70.4) | 99.8 (86.2) | 99.9 (73.7) | 99.5 (84.2) |
| Molecules/ asymmetric unit | 1 | 1 | 1 | 2 |
| Refinement | | | | |
| $R_{work}$†/$R_{free}$‡ (%) | 18.19/18.91 | 16.57/18.39 | 19.87/23.54 | 18.39/21.98 |
| No. of atoms | 1955 | 1861 | 1732 | 15,595 |
| No. of residues | 1600 | 1573 | 1580 | 15,256 |
| No. of water molecules | 343 | 273 | 87 | 96 |
| No. of ligands | 12 | 15 | 67 | 243 |
| RMSD from ideality | | | | |
| Bond length (Å) | 0.005 | 0.006 | 0.008 | 0.004 |
| Bond angle (°) | 0.84 | 0.84 | 0.91 | 0.65 |
| **Ramachandran plot (%)** | | | | |
| Favored | 99.02 | 98.54 | 98.52 | 96.16 |
| Allowed | 0.98 | 1.46 | 1.48 | 3.74 |
| Outliers | 0.00 | 0.00 | 0.00 | 0.11 |
| PDB accession code | 8IN3 | 8IN4 | 8IN6 | 8IN1 |

The highest resolution shell is shown in parentheses.

*$R_{meas} = \Sigma_{hkl}\{N(hkl)/[N(hkl) - 1]\}^{1/2} \Sigma_i|I_i(hkl) - <I(hkl)> |/ \Sigma_{hkl} \Sigma_i|I_i(hkl)$, where $I_i(hkl)$ is the *i*th observation of the intensity of reflection *hkl* and $\langle I(hkl)\rangle$ is the mean over *n* observations.

†$R_{work} = \Sigma_{hkl}||F_{obs}(hkl)| - |F_{calc}(hkl)||/ \Sigma_{hkl}|F_{obs}(hkl)|$.

‡$R_{free}$ was calculated with an approximate 5% fraction of randomly selected reflections evaluated from refinement.

loops, LL1 (Q80–N91) and LL2 (R147– G163), and arranged in a triangular shape (*Figure 2A*). These three PADs share a similar structure, with a root-mean-squared difference (RMSD) of 1.065 Å over 46 Cα atoms and only ~20.3% sequence identity (*Figure 2B, C*). The three PADs share a canonical CBM14 fold consisting of two β-sheets containing three N-terminal and two C-terminal antiparallel β-strands. Additionally, two small α-helices were appended to the N- and C-terminus in PAD1 and PAD3 but not in PAD2 (*Figure 2B*).

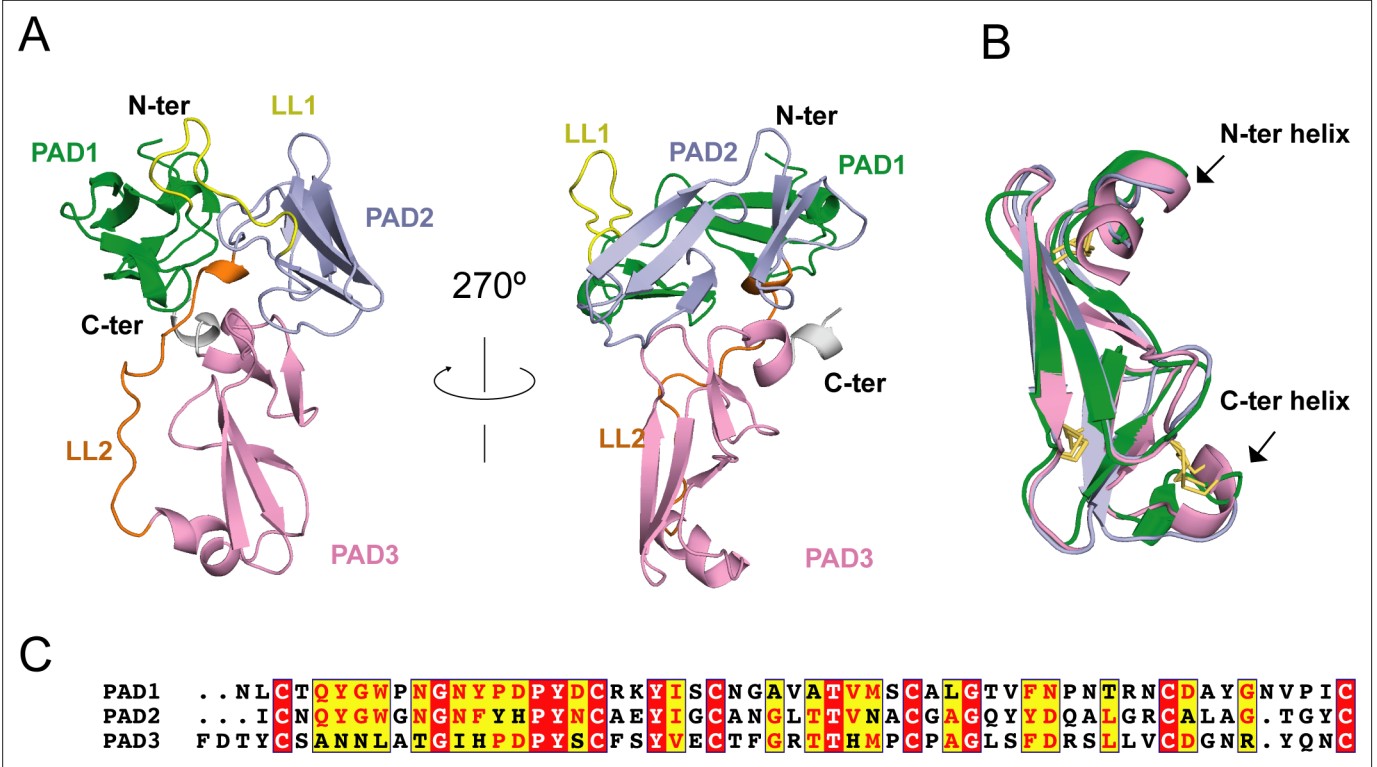

**Figure 2.** *Eisenia* hydrolysis-enhancing protein (EHEP) structure. (**A**) Cartoon representation of EHEP. The three peritrophin-A domains (PADs) are colored green, light blue, and pink, respectively. Linker long loop1 (LL1) and loop2 (LL2) are colored yellow and blue. (**B**) Structural superposition of the three PAD domains of EHEP. The three domains are colored as in (**A**). The disulfide bonds are shown as yellow sticks. (**C**) Sequence alignment of three PAD domains. Alignment was performed by CLUSTALW and displayed with ESPript3.

The online version of this article includes the following source data and figure supplement(s) for figure 2:

**Figure supplement 1.** Acetylation modification of *Eisenia* hydrolysis-enhancing protein (EHEP).

**Figure supplement 1—source data 1.** recomEHEP and *Eisenia* hydrolysis-enhancing protein (EHEP) activity.

Although the Dali server (*Holm and Rosenström, 2010*) did not provide similar structures using the overall structure of EHEP as the search model, six structures showed similarities with a single PAD of EHEP. These structures were the members of the PAD family, including the chitin-binding domain of chitotriosidase (PDB ID 5HBF) (*Fadel et al., 2016*), avirulence protein 4 from *Pseudocercospora fuligena* [*Pf*Avr4 (PDB ID 4Z4A)] (*Kohler et al., 2016*) and *Cladosporium fulvum* [*Cf*Avr4 (PDBID 6BN0)] (*Hurlburt et al., 2018*), allergen Der p 23 (PDB ID 4ZCE) (*Mueller et al., 2016*), tachytitin (PDB ID 1DQC) (*Suetake et al., 2000*), and allergen Blot 12 (PDB ID 2MFK), with Z-scores of 4.7–8.4, RMSD values of 1.2–2.8 Å, and sequence identity of 19–37%. The highest sequence disparity was detected in PAD2, whereas the greatest structural differences were noted in PAD3. The $C^{No1}X_{15}C^{No2}X_5C^{No3}X_9C^{No4}X_{12}C^{No5}X_{5-9}C^{No6}$ motif (superscripts and subscripts indicate the cysteine number and number of residues between adjacent cysteines, respectively) in each PAD of EHEP formed three disulfide bonds between the following pairs: $C^{No1}$–$C^{No3}$, $C^{No2}$–$C^{No6}$, and $C^{No4}$–$C^{No5}$ (*Figure 2B*). Such rich disulfide bonds may play a folding role in the structural formation of EHEP, with >70% of the backbone in a loop conformation. A similar motif with disulfide bonds was observed in tachycitin (*Suetake et al., 2000*), *Pf*Avr4 (*Kohler et al., 2016*), *Cf*Avr4 (*Hurlburt et al., 2018*), and the chitin-binding domain of chitinase (*Fadel et al., 2016*). Although these proteins share a highly conserved core structure, they have different biochemical characteristics. For example, the chitin-binding domain of human chitotriosidase, Avr4, and tachycitin possess chitin-binding activity, but the critical residues for chitin binding are not conserved (*Fadel et al., 2016*; *Hurlburt et al., 2018*; *Madland et al., 2019*), indicating that they employ different binding mechanisms. In contrast, EHEP and allergen Der p 23 do not possess chitin-binding activity (*Tsuji et al., 2017*; *Mueller et al., 2016*). Thus, the PAD family may participate in several biochemical functions.

## Modification of EHEP

Consistent with the molecular weight results obtained using MALDI–TOF MS (*Sun et al., 2020*), the apo structure2 (1.4 Å resolution) clearly showed that the cleaved N-terminus of Ala21 underwent acetylation (*Figure 2—figure supplement 1A*), demonstrating that EHEP is acetylated in *A. kurodai* digestive fluid. N-terminal acetylation is a common modification in eukaryotic proteins. Such acetylation is associated with various biological functions, such as protein half-life regulation, protein secretion, protein–protein interaction, protein–lipid interaction (*Silva and Martinho, 2015*), metabolism, and apoptosis (*Hollebeke et al., 2012*). Furthermore, N-terminal acetylation may stabilize proteins (*Lange and Overall, 2011*). To explore whether acetylation affects the protective effects of EHEP on *aku*BGL, we used the *E. coli* expression system to obtain the unmodified recomEHEP (A21–K229). We measured the TNA-precipitating assay of recomEHEP. The results revealed that recomEHEP precipitated after incubation with TNA at a comparable level to that of natural EHEP (*Figure 2—figure supplement 1B*), indicating that acetylation is not indispensable for the phlorotannin-binding activity and stabilization of EHEP. Future studies are warranted to verify the exact role of N-terminal acetylation of EHEP in *A. kurodai*.

## TNA binding to EHEP

To understand the mechanism by which TNA binds to EHEP, we determined the structure of EHEP complexed with TNA (EHEP–TNA) using the soaking method. In the obtained structure, both $2F_o–F_c$ and $F_o–F_c$ maps showed the electron density of 1,2,3,4,6-pentagalloylglucose, a core part of TNA missing the five external gallic acids (*Figure 3A*, *Figure 3—figure supplement 1*). Previous studies have shown that acid catalytic hydrolysis of TNA requires a high temperature of 130°C (*Jie Fu et al., 2015*); even with a polystyrene-hollow sphere catalyst, a temperature of 80°C is required (*Luo et al., 2018*). Therefore, the five gallic acids could not be visualized in the EHEP–TNA structure, most likely due to the structural flexibility of TNA.

The apo EHEP and EHEP–TNA structures were highly similar, with an RMSD value of 0.283 Å for 207 Cα atoms (*Figure 3—figure supplement 1B*). However, the superposition of the two structures showed a decrease in the loop part of EHEP–TNA. TNA binding caused a slight increase in the α-helix and β-sheet contents of PAD2 and PAD3 (*Figure 3—figure supplement 1B*). In the EHEP–TNA structure, the residues C93–Y96 of PAD2 folded into an α-helix and each β-sheet of the first β-strand in PAD3 elongated by incorporating one residue in the first ($G^{176}$) and second β-sheets ($S^{186}$) and three residues in the third β-sheet ($H^{197}MP^{199}$).

The EHEP–TNA structure revealed that TNA binds to the center of the triangle formed by the three PADs, a positively charged surface (*Figure 3A* and *Figure 3—figure supplement 1C*). The binding pocket on the EHEP surface was formed by the C-terminal α-helix of PAD1, the N-terminal α-helix of PAD2, and the middle part (loop) of PAD3 assisted by two long linker loops (LL1 and LL2). TNA was primarily bound to EHEP via hydrogen bonds and hydrophobic interactions (*Figure 3B*). Gallic acid1, 4, and 6 interacted with EHEP via hydrogen bonds and additional hydrophobic contacts, whereas gallic acid2 and 3 only interacted hydrophobically with EHEP. The 3-hydroxyl groups of gallic acid1 and 6 individually formed a hydrogen bond with the main chain of G74 and the side chain of N75 in PAD1. The main chain of Y96 and P199 in PAD2 and PAD3 formed hydrogen bonds with the gallic acid4. Additionally, some hydrogen bonds were formed between TNA and water molecules. TNA binding was also stabilized by hydrophobic interactions between the benzene rings of gallic acid and EHEP. For instance, gallic acid4 and 6 are stacked with P201 and P77, respectively; moreover, gallic acid3 and 4 are stacked with P199.

The EHEP–TNA structure clearly showed that TNA binds to EHEP without covalent bonds, and the binding does not induce significant structural changes; thus, we attempted to recover EHEP from EHEP–TNA precipitates by adjusting the pH. As hypothesized, resolubilization of the EHEP–phlorotannin precipitate is pH dependent (*Figure 3C*). The EHEP–TNA precipitate did not resolubilize at pH 7.0; however, after incubating for >1 hr at pH 7.5, the precipitate started resolubilizing. Most of the precipitate rapidly resolubilized at an alkaline pH (≥8.0) after incubation for 15 min. Furthermore, the resolubilized EHEP had the same elution profile as that of the natural EHEP (*Figure 3D*) in SEC, suggesting that resolubilized EHEP maintained the native structure and its phlorotannin-precipitate activity (*Figure 3—figure supplement 1D*).

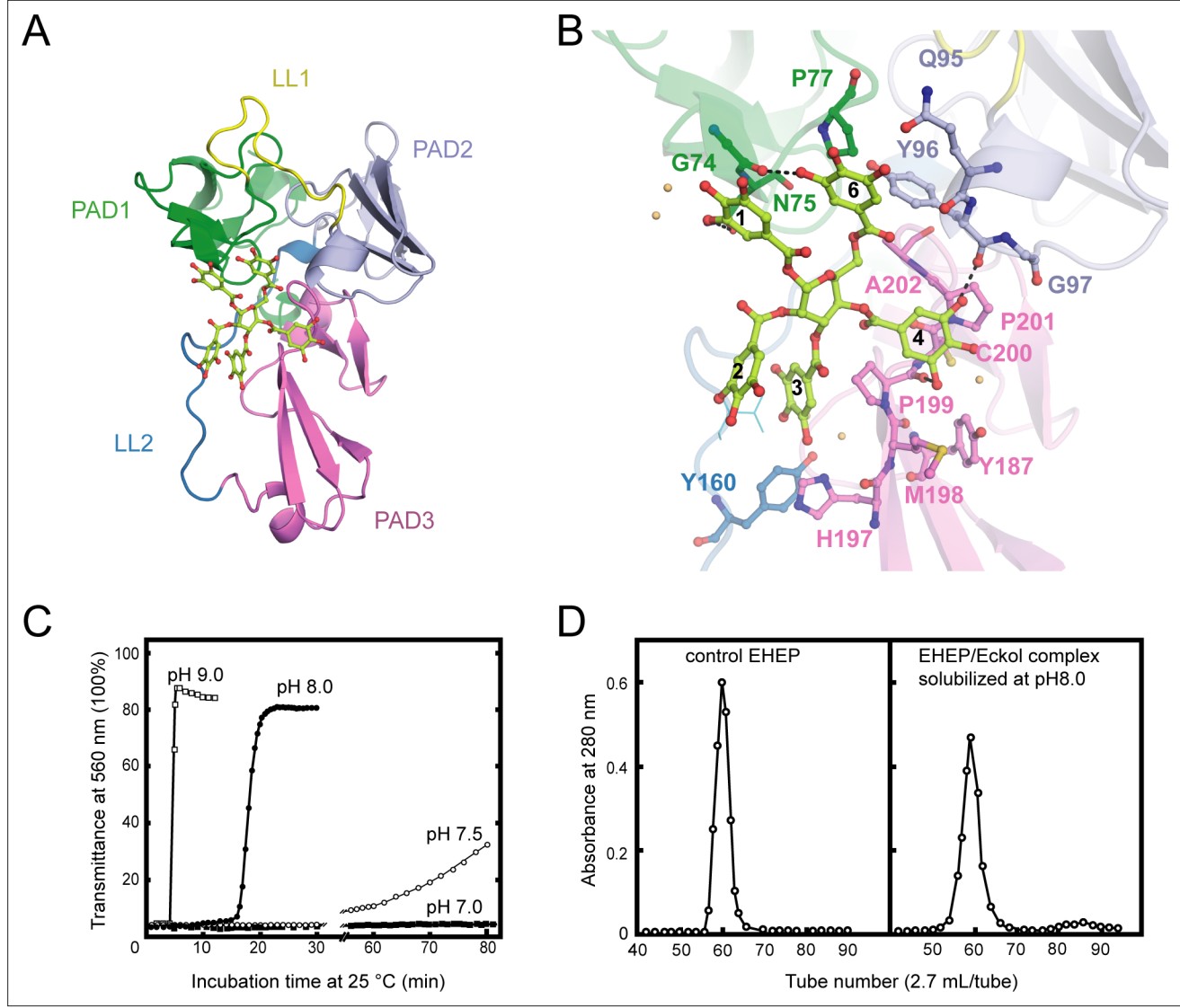

**Figure 3.** Structure of *Eisenia* hydrolysis-enhancing protein (EHEP)–tannic acid (TNA). (**A**) The overall structure of EHEP–TNA. (**A**) The overall structure of EHEP–TNA. EHEP and TNA are shown by the cartoon and stick model, respectively. EHEP is colored as in *Figure 1*. The C and O atoms of TNA are colored lemon and red, respectively. (**B**) Interaction of TNA (ball-stick in the same color as (**A**)) with EHEP (cartoon in the same color as (**A**)) in EHEP–TNA structure. The residues of EHEP in contact are labeled and shown by a ball-stick with N, O, and S atoms in blue, red, and brown, respectively. The C and O atoms of TNA are colored the same as (**A**), lemon, and red, respectively. Dashed lines show hydrogen bonds. The water molecules stabilizing TNA were shown as light orange spheres. (**C**) Effect of pH on resolubilization of an EHEP–eckol precipitate. Buffers with pH 9.0, 8.0, 7.5, and 7.0 are presented as hollow square, solid circle, hollow circle, and solid square, respectively. (**D**) The EHEP–eckol precipitate was dissolved in 50 mM Tris–HCl (pH 8.0) and analyzed using a gel filtration column of Sephacryl S-100.

The online version of this article includes the following source data and figure supplement(s) for figure 3:

**Figure supplement 1.** *Eisenia* hydrolysis-enhancing protein (EHEP)–tannic acid (TNA) structure and structural comparisons.

**Figure supplement 1—source data 1.** Resolubilized *Eisenia* hydrolysis-enhancing protein (EHEP) activity.

## Two domains of *aku*BGL

To reveal the structural basis of *aku*BGL recognition of laminarin and its inhibition by TNA, we attempted to determine its structure. We soaked crystals in TNA as well as various substrate solutions but finally obtained the optimal resolution using crystal soaking in TNA. There was no electron density of TNA or something similar in the $2F_o–F_c$ and $F_o–F_c$ map of the obtained structure; thus, we considered this structure as the apo form of *aku*BGL.

Two *aku*BGL molecules were observed in an asymmetric unit (MolA and MolB). These molecules lacked the N-terminal 25 residues (M1–D25), as confirmed by N-terminal sequencing analysis of purified natural *aku*BGL. This N-terminal fragment was predicted to be a signal peptide using the web server SignalP-5.0. The residues L26–P978 were constructed in MolA and MolB with glycosylation, whereas the remaining C-terminal residues (A979–M994) could not be visualized as they were disordered. The electron density map of $F_o$–$F_c$ revealed *N*-glycosylation at three residues, that is, N113, N212, and N645 (*Figure 4—figure supplement 1A, B*). *N*-glycosylation of GH enzymes prevents proteolysis and increases thermal stability (*Amore et al., 2017*; *Han et al., 2020b*). Additionally, a study on β-glucosidase *Aspergillus terreus* BGL demonstrated that *N*-glycosylation of N224 affected the folding stability, even when it is located close to a catalytic residue (*Wei et al., 2013*). In *aku*BGL, all *N*-glycosylation sites were present on the surface, far from the catalytic site. Therefore, we speculate that *aku*BGL glycosylation does not affect its activity. Except for the difference in visualized

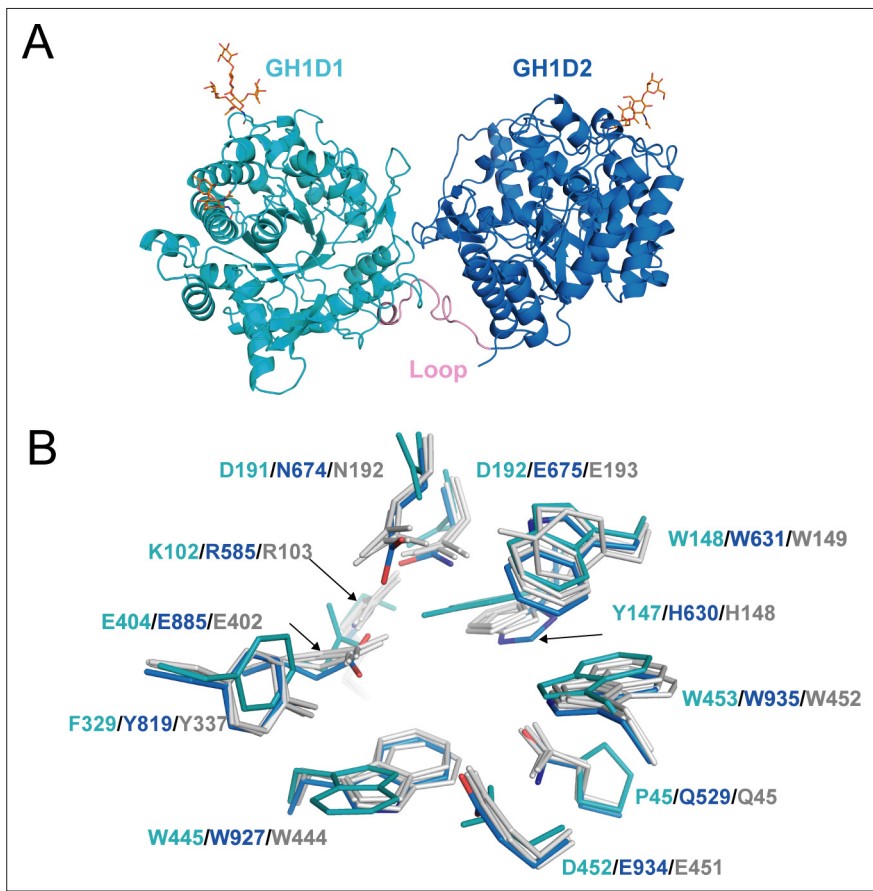

**Figure 4.** Structure of *aku*BGL. (**A**) Overall structure. The GH1D1 (light blue) and GH1D2 (cyan) domains are linked by a long loop (linker loop) colored in pink. The *N*-linked glycans were shown in the orange stick. (**B**) Residues superposition of the glycone-binding site (GBS) and catalysis-related residue (CR) site of the domains GH1D1 (cyan), GH1D2 (light blue) with β-glucosidase structures from termite *Neotermes koshunensis* (*Nk*BGL, gray PDB ID 3VIH) (*Jeng et al., 2012*), β-glucosidase from rice (*Os*BGL, gray, PDB ID 2RGL) (*Chuenchor et al., 2008*), and β-glucosidase from *Bacillus circulans* sp. *Alkalophilus* (gray, PDB ID 1QOX) (*Hakulinen et al., 2000*). Only the residue numbers of GH1D1 (cyan), GH1D2 (light blue), and *Nk*BGL (gray) are shown for clarity.

The online version of this article includes the following source data and figure supplement(s) for figure 4:

**Figure supplement 1.** *aku*BGL structure.

**Figure supplement 1—source data 1.** Sodium dodecyl sulfate–polyacrylamide gel electrophoresis (SDS–PAGE) of recomGH1D1.

**Figure supplement 2.** Structural comparison of BGLs.

**Figure supplement 3.** The model of GH1D2 docking with the substrate laminaritetraose.

glycans resulting from glycosylation, MolA and MolB were similar, with an RMSD value of 0.182 for 899 Cα atoms; therefore, we used MolA for further descriptions and calculations.

The structure of *aku*BGL consisted of two independent GH1 domains, GH1D1 (L26–T494) and GH1D2 (D513–P978), linked by a long loop (D495–Y512) (**Figure 4A**). There was little interaction between GH1D1 and GH1D2, only in a buried surface area comprising 2% of the total surface (708.9 Å²) (**Figure 4—figure supplement 1C**). GH1D1 and GH1D2 have a sequence identity of 40.47% and exhibit high structural similarity with an RMSD value of 0.59 Å for 371 Cα atoms (**Figure 4—figure supplement 2A**, upper panel).

Glucosidases of the GH1 family utilize a retaining mechanism with two glutamic acids to catalyze glucoside hydrolysis. In general, the distance between the two catalytic oxygen atoms of the side chain of two glutamic acids is approximately 5 Å (**Hayashi et al., 2007**). Sequence and structure alignment of GH1D1 and GH1D2 of *aku*BGL with other members of the GH1 family revealed that the second glutamate is conserved (E404), but the first glutamate is replaced by D192 in GH1D1. The oxygen atoms of the side chains between D192 and E404 of GHD1 were 8.4 Å apart. In contrast, GH1D2 conserved two glutamic acids (E675 and E885) at the carboxyl termini of β-strands 4 and 7; the distance between the oxygen atoms of the E675 and E885 side chains was 5.1 Å (**Figure 4—figure supplement 2A** bottom panel), similar to that of *Neotermes koshunensis* BGL (*Nk*BGL, PDB ID 3VIH) (**Jeng et al., 2012**), *Nannochloropsis oceanica* BGL (*No*BGL, PDB ID 5YJ7) (**Dong et al., 2021**), and *Spodoptera frugiperda* BGL (PDB ID 5CG0) (3.9–4.9 Å) (**Tamaki et al., 2016**). Furthermore, regarding the two other conserved essential regions for β-glucosidase activity, namely, the glycone-binding site (GBS) and catalysis-related residues (CR), GH1D1 conserved neither GBS nor CR, whereas GH1D2 conserved both (**Figure 4B**). Altogether, we suggest that GH1D1 does not possess catalytic activity. We expressed and purified the recombinant GH1D1, which did not show any hydrolytic activity toward O-PNG (**Figure 4—figure supplement 1D, E**), although we could not rule out the effect of *N*-glycosylation.

Structural comparison of GH1D2 with other BGLs, including *Nk*BGL (PDB ID 3VIH) (**Jeng et al., 2012**), rice (*Oryza sativa L.*) BGL (*Os*BGL, PDB ID 2RGL) (**Chuenchor et al., 2008**), and microalgae *No*BGL (PDB ID 5YJ7) (**Dong et al., 2021**), revealed the characteristics of each active pocket (**Figure 4—figure supplement 2B**). *Os*BGL and *No*BGL have deep, narrow, and straight pockets, whereas GH1D2 and *Nk*BGL have broad and crooked pockets. Such active pocket shapes reflect the substrate preferences of *Os*BGL and *No*BGL; they hydrolyze laminaribiose with no detectable activity toward laminaritraose (**Dong et al., 2021**; **Opassiri et al., 2003**). Furthermore, the difference in the features of large active pockets between *Nk*BGL and GH1D2, wherein GH1D2 often possesses an auxiliary site with several aromatic residues bound to the carbohydrate via CH–π interactions (**Hudson et al., 2015**), may explain their substrate specificity. *Nk*BGL efficiently hydrolyzes laminaribiose and cellobiose but has weak hydrolytic activity toward laminarin (**Ni et al., 2007**). In contrast, the GH1D2 of *aku*BGL has similar activity levels toward cellobiose and laminarin (**Tsuji et al., 2013**). Therefore, the GH1D2 of *aku*BGL may recognize larger substrates than that of other BGLs. Laminarin typically has a curved conformation; accordingly, narrow- and straight-shaped pockets are incompatible for binding. Furthermore, we docked GH1D2 with laminaritraose, wherein the four glucose units formed extensive contacts with GH1D2. Hydrogen bonds are involved in the catalytic residues E675. In addition, several aromatic residues, such as W631, F677, W681, F689, Y819, Y846, W857, and W935, formed CH–π stacking interactions (**Figure 4—figure supplement 3**). Some interacting residues belonged to GBS and CR sites, such as E675, W631, Y819, and W935. Additionally, the docking structure revealed that the +3 and +4 glucose of laminaritraose are located at the auxiliary binding site and that atom O1 of the +4 glucose is positioned outside the pocket (**Figure 4—figure supplement 3**), implying that the auxiliary binding site with several aromatic residues (F677 and W681) of GH1D2 facilitates laminarin binding.

## Inhibitor binding of *aku*BGL

As we could not obtain the complex structure of *aku*BGL with TNA, we performed docking calculations of *aku*BGL GH1D2 with TNA to explore the inhibition mechanism. The docking model of *aku*BGL–TNA showed that seven gallic acid rings of TNA formed an extensive hydrogen bond network with *aku*BGL in the binding pocket (**Figure 5**). The hydroxyl groups of TNA formed hydrogen bonds with the residues N552, E675, D735, K739, K759, Q840, T844, D852, and K859 of GH1D2. Moreover,

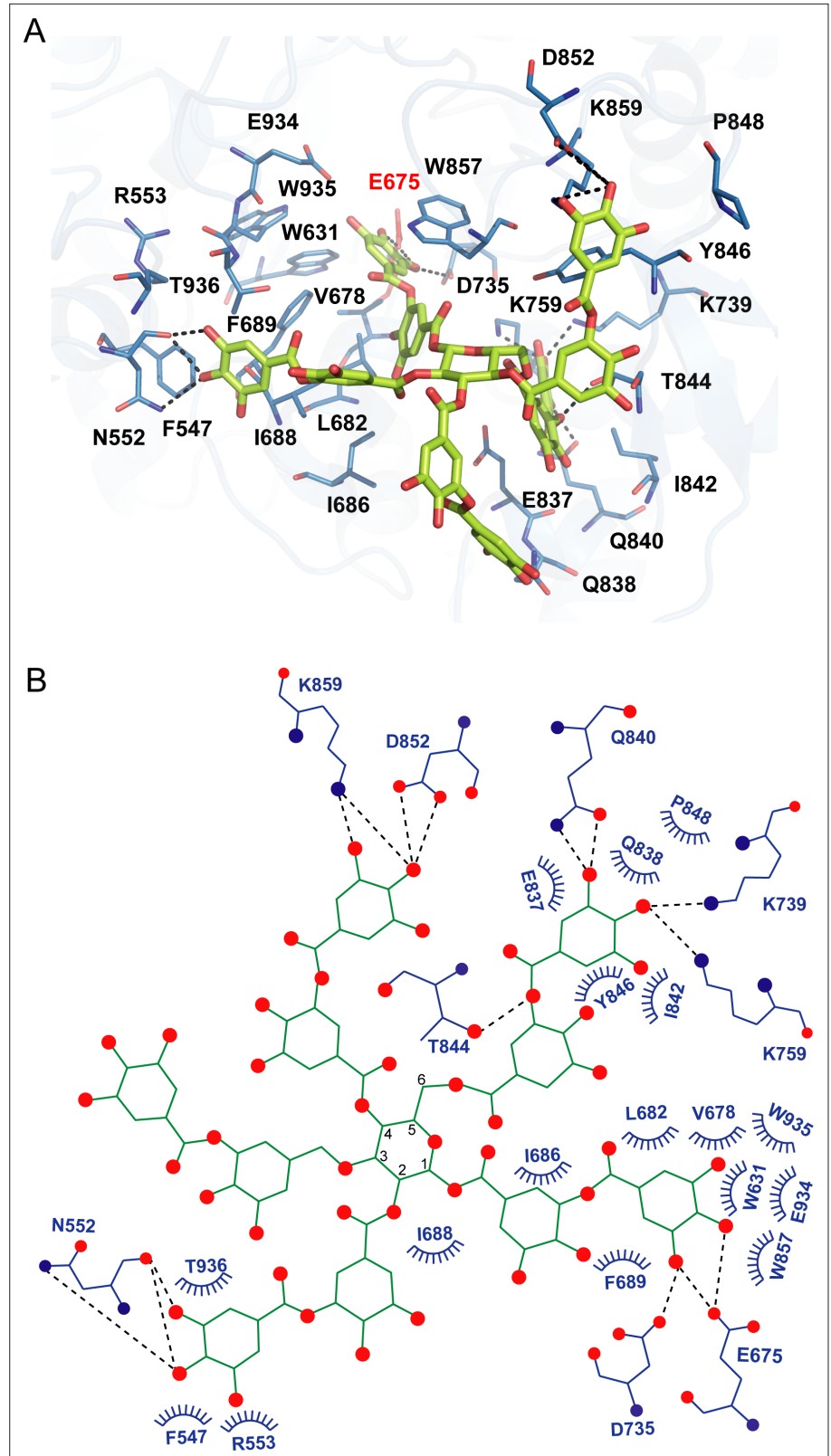

**Figure 5.** Docking model of *aku*BGL with tannic acid (TNA). (**A**) Detailed interaction between *aku*BGL and TNA in the docking model. TNA is shown in the green stick model. The hydrogen bonds are shown as dashed lines. (**B**) A 2D diagram of the interaction between *aku*BGL and TNA shown in (**A**). The hydrogen bonds are shown as dashed lines, and the hydrophobic contacts as circular arcs.

*Figure 5 continued on next page*

*Figure 5 continued*

The online version of this article includes the following figure supplement(s) for figure 5:

**Figure supplement 1.** The models of GH1D2 docking.

---

benzene rings showed hydrophobic interactions with several hydrophobic residues. In particular, stable π–π stacking was observed between TNA and residues F547, W631, F689, Y846, W857, and W935. Among these residues, the conserved E675 was the catalytic residue, and W631, W935, and E934 contributed to GBS and CR sites.

In addition, we performed a docking calculation of GH1D2 with the characteristic inhibitors eckol and phloroglucinol (*Jung et al., 2010*). The binding mechanisms of eckol and phloroglucinol were similar to those of TNA but with different contact residues (*Figure 5—figure supplement 1*). For eckol, the six hydroxyl groups formed hydrogen bonds with residues E675, D735, E737, K759, E885, and E934. Additionally, residues W631, F677, F689, Y819, W857, W935, F943, and W927 formed π–π stacking interactions with eckol. For phloroglucinol, the three hydroxyl groups formed hydrogen bonds with E675, E885, and E934, whereas residues W631, F689, Y819, W857, W935, and W927 formed π–π stacking interactions with the benzene ring.

The docking scores of the inhibitors TNA, eckol, and phloroglucinol were −8.8, −7.3, and −5.7 kcal/mol, respectively, whereas the substrate laminaritetraose had a docking score of −6.6 kcal/mol. The docking scores corroborated well with the inhibition activity toward *aku*BGL, that TNA had a more robust inhibition activity than phloroglucinol (*Tsuji et al., 2017*), indicating that the docking results are reasonable. In summary, the three inhibitors interacted with *aku*BGL through similar binding mechanisms to occupy the substrate-binding site, suggesting a reversible competitive inhibition mechanism.

## Discussion

In marine habitats, the ecological interactions between brown algae and herbivores dominate marine ecosystems (*Amsler and Fairhead, 2005*). The *aku*BGL–phlorotannin/laminarin–EHEP system represents the feeding defense–offense associations between *A. kurodai* and brown algae. We focused on this system to understand the molecular mechanism at the atomic level. In contrast to most GH1 BGLs containing one catalytic GH1 domain, *aku*BGL consists of a noncatalytic GH1D1 and a catalytic GH1D2. The noncatalytic GH1D1 may act as a chaperone for GH1D2, as we successfully overexpressed GH1D1 but failed to do the same for GH1D2. Such multi-GH1D assembly and a similar function have been suggested in β-glucosidase *Cj*CEL1A of *Corbicula japonica* (*Sakamoto et al., 2009*) and glycosidase *Lp*MDGH1 of the shipworm *Lyrodus pedicellatus* (*Sabbadin et al., 2018*). *Cj*CEL1A has two tandem GH1Ds with a sequence identity of 43.41% (*Sakamoto et al., 2009*). Two catalytic glutamic acids and the residues related to substrate binding are conserved in the second GH1D, whereas the first GH1 domain lacks these conserved residues and may play a role in folding the catalytic domain. *Lp*MDGH1 consists of six GH1Ds, among which GH1D2, 4, 5, and 6 contain the conserved residues for activity, whereas others do not contain these residues and might be involved in protein folding or substrate interactions (*Sabbadin et al., 2018*). Future assays of GH1D2 and its inactive mutants are the complement to validate the molecular mechanism of *aku*BGL.

BGLs have different substrate preferences in the degree of polymerization and type of glycosidic bond. In general, BGLs prefer to react with mono-oligo sugars over polysaccharides. For instance, *Os*BGL, *No*BGL, and *Nk*BGL hydrolyze disaccharides (cellobiose and laminaribiose) but display no or weak activity toward polysaccharides (cellulose and laminarin) (*Dong et al., 2021*; *Opassiri et al., 2003*; *Ni et al., 2007*). The structure of GH1D2 explained the substrate preference for the polysaccharide laminarin. GH1D2 contains an additional auxiliary site composed of aromatic residues (*Figure 4—figure supplement 2B*) in the substrate entrance pocket, which putatively enables it to accommodate a long substrate, contributing to *aku*BGL activity toward laminarin, as supported by docking calculations (*Figure 4—figure supplement 3*). In addition, docking analysis of *aku*BGL GH1D2 with inhibitors (TNA, phlorotannin, eckol, and phloroglucinol) revealed that these inhibitors bind to the substrate-binding site via hydrogen bonds and hydrophobic interactions similar to laminarin. Such binding mechanisms suggest the presence of competitive inhibition to occupy the binding site, consistent with previous research (*Tsuji et al., 2017*). Future kinetic experiments are required to quantitatively validate the competitive inhibition of phlorotannin against *aku*BGL.

EHEP, expressed in the midgut of *A. kurodai*, was identified as an antidefense protein, protecting the hydrolysis activity of *aku*BGL from phlorotannin inhibition (*Tsuji et al., 2017*). Such an ecological balance also exists between plants and their predatory mammals and insects. Similar to brown algae, plants use the toxic secondary metabolite tannins as their defense mechanism against predators, constituting 5%–10% of the dry weight of leaves. In vertebrate herbivores, tannins reduce protein digestion (*Barbehenn and Peter Constabel, 2011*). In phytophagous insects, tannins may be oxidized in the alkaline pH of the insect midgut and cause damage to cells (*Marsh et al., 2020*). The evolution of plant–herbivore survival competition has led to the development of remarkably unique adaptation strategies. Mammals feeding on plants that contain tannin may overcome this defense by producing tannin-binding proteins, proline-rich proteins, and histatins (*Shimada, 2006*; *De Smet and Contreras, 2005*). Proline constitutes at least 20% of the total amino acid content in proline-rich proteins; for some species, the proportion of proline reaches 40%. Histidine constitutes 25% of the total amino acid content in histatins. Both proline-rich proteins and histatins are unfolded proteins with random coils in solution. In caterpillars, the oxidation damage of tannin is reduced by the low oxygen level. Some insects use the peritrophic membrane to transport tannins into the hemolymph, where they are excreted (*War et al., 2020*). Additionally, the peritrophic envelop protects insects from tannins by forming an impermeable barrier to tannins (*Barbehenn and Peter Constabel, 2011*). *A. kurodai* uses a similar strategy to mammals by secreting the tannin-binding protein EHEP. Although EHEP has a completely different amino acid composition with proline-rich proteins and histatins, EHEP also binds to phlorotannin. Therefore, EHEP may be a specific counteradaptation that allows *A. kurodai* to feed on brown algae, as there are no homologous proteins in other organisms.

The three PADs of EHEP are arranged in a triangular shape, forming a large cavity on the surface at the triangle center to provide a ligand-binding site. EHEP has a positively charged surface at a pH of <6.0, whereas the surface becomes negatively charged at a pH of >7.0 (*Figure 3—figure supplement 1C*). Meanwhile, TNA has a pKa of 4.9–8 (*Lin et al., 2009*; *Ge et al., 2019*; *Yi et al., 2011*), showing minor negative charges at an acidic pH and the highest negative charge at a pH of >7.0 (*Dultz et al., 2021*). Therefore, TNA binds to EHEP at a pH of <6.0 (pH of crystallization = 4.5) but shows charge repulsion with EHEP at a pH of >8.0. Altogether, TNA is protonated and behaves as a hydrogen bond donor when the pH is below its pKa, whereas when the pH is above its pKa, TNA is deprotonated and the hydrogen bonding cannot be maintained. As losing hydrogen bonds and increasing repulsive forces at a pH >8.0, the precipitated EHEP–TNA could dissolve in the buffer of pH >8.0. This pH-induced reversible interaction also occurred in other proteins, such as BSA, pepsin, and cytochrome C (*Han et al., 2020a*). The phlorotannin members share a similar structure with TNA; thus, we speculate that the EHEP–phlorotannin complex also exhibits a pH-induced reversible interaction. In vivo, the pH of the digestive fluid of *A. kurodai* is approximately 5.5 (*Tsuji et al., 2017*), which favors the binding of EHEP to phlorotannin. In the alkaline hindgut (*Lemke et al., 2003*), the EHEP–phlorotannin disassociates (*Figure 6*), and the phlorotannin is subsequently excreted from the anus.

Based on the EHEP–TNA structure and docking models of *aku*BGL–inhibitor/substrate, we proposed a mechanism of phlorotannin inhibition on *aku*BGL activity and EHEP protection from phlorotannin inhibition (*Figure 6*). Because laminarin lacks the benzene rings essentially to form CH–π stacking interactions with EHEP, the EHEP can be considered not bind with laminarin. In the absence of EHEP, phlorotannin occupies the substrate-binding site of *aku*BGL, inhibiting the substrate from entering the active site and resulting in no glucose production. In the presence of EHEP, it competitively binds to phlorotannin, freeing the *aku*BGL pocket. Then, the substrate can enter the active pocket of *aku*BGL, and glucose can be produced normally. The digestive fluid of *A. kurodai* contains EHEP at a high concentration (>4.4 µM) (*Tsuji et al., 2017*), which is slightly higher than the concentration of EHEP (3.36 µM) that protects *aku*BGL activity (*Figure 1B*). The high concentration of EHEP allows *A. kurodai* to feed on phlorotannin-rich brown algae. The balance between phlorotannin inhibition and protection is controlled by the concentrations of phlorotannin and EHEP in vivo. The kinetic analysis of the *aku*BGL–phlorotannin/laminarin–EHEP system will provide detailed reaction parameters.

The *aku*BGL–phlorotannin/laminarin–EHEP system represents the digestive–defensive–offensive associations between algae and herbivores. Our study presented the molecular mechanism of this system at the atomic level, providing a molecular explanation for how the sea hare *A. kurodai* utilizes EHEP to protect *aku*BGL activity from phlorotannin inhibition. Furthermore, such a feeding strategy

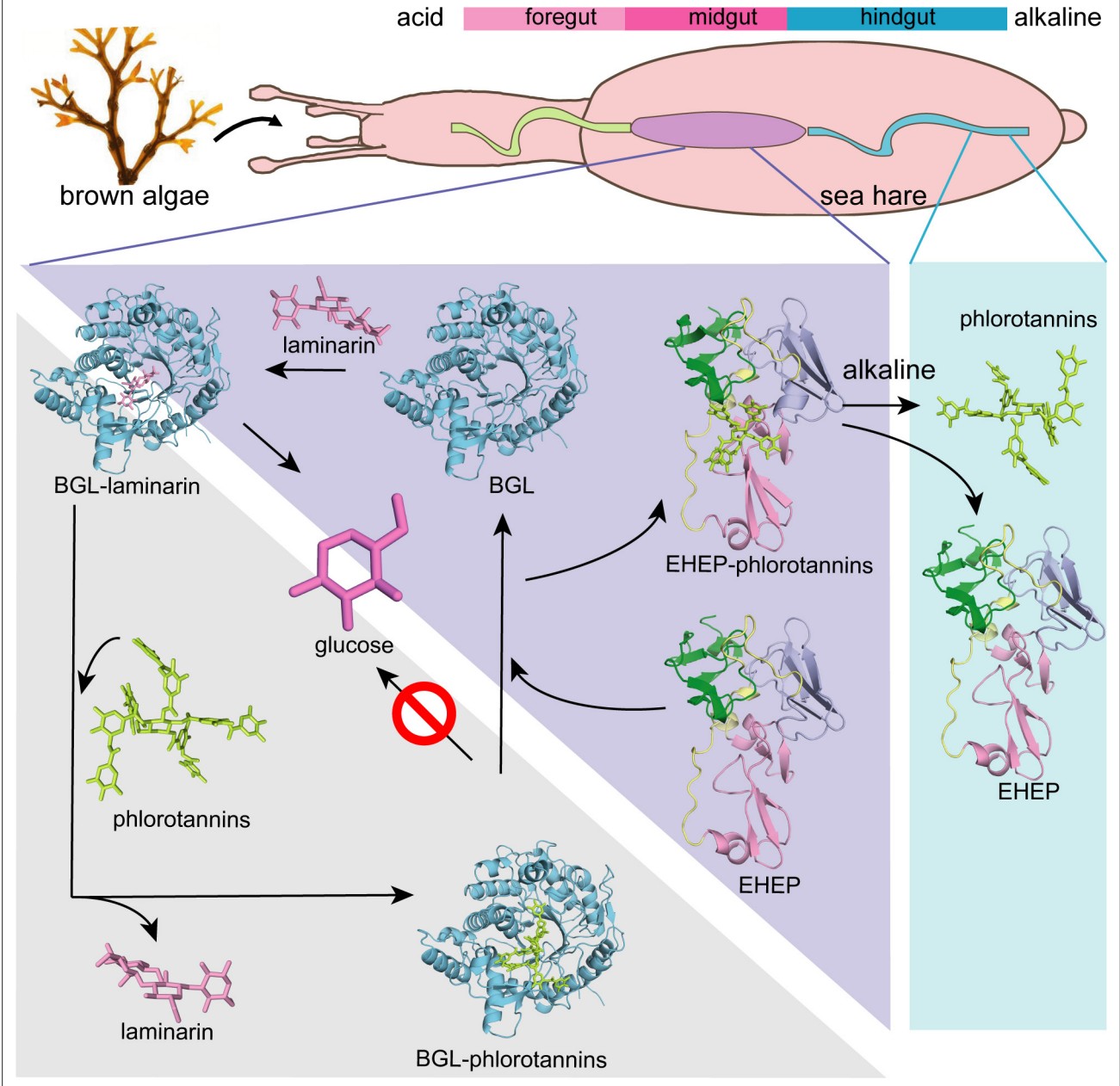

**Figure 6.** Proposed molecular mechanisms. Proposed molecular mechanisms of tannic acid (TNA) inhibition of *aku*BGL activity and *Eisenia* hydrolysis-enhancing protein's (EHEP) protective effects of *aku*BGL in *aku*BGL–phlorotannin/laminarin–EHEP system (light purple triangle). The digestive tract of *A.kurodai* consists of foregut (blue), midgut (purple), and hindgut (blue) (top). The bar chart above depicts the pH of the digestive tract, with pink denoting acid and blue denoting alkalinity.

has attracted attention for producing glucose as a renewable biofuel source, so our studies provide a molecular basis for the biofuel industry applications of brown algae.

## Materials and methods
### EHEP and *aku*BGL preparation

Natural EHEP (22.5 kDa) and *aku*BGL (110 kDa) were purified from *A. kurodai* digestive fluid as previously described (*Tsuji et al., 2017*). For crystallization, we added one step for purification of EHEP by size-exclusion chromatography using a HiLoad 16/60 Superdex 75 column (GE, America) equilibrated

in 20 mM MES [2-(4-morpholino) ethanesulfonic acid]–NaOH buffer (pH 6.5). Collected EHEP was then concentrated to 15–25 mg/ml using Vivaspin-4 10K columns (Sartorius, Göttingen, Germany). For the *aku*BGL, we exchanged the buffer from 20 mM Tris–HCl pH 7.0–20 mM Bis-Tris pH 6.0 with Amicon-Ultra (molecular weight cut-off of 50 kDa) and concentrated to 11 mg/ml.

To verify whether the chemical modification indicated by the previous study (*Sun et al., 2020*) affects the function of EHEP, we prepared recombinant EHEP (recomEHEP) without the N-terminal signal peptide (1–20 aa) and the chemical modification (*Sun et al., 2020*). EHEP cDNA was obtained via reverse transcription-polymerase chain reaction using the total RNA of *A. kurodai* as the template. The reamplified fragment was digested and ligated to a plasmid derived from pET28a (Novagen, Darmstadt, Germany). The primers used were shown in *Supplementary file 1*. The resulting plasmid encoding recomEHEP with an N-terminal hexahistidine-tag was transformed into *E. coli* B834(DE3) pARE2 cells. The cells were cultured in lysogeny broth (LB) medium with the antibiotics kanamycin (25 mg/l) and chloramphenicol (34 mg/l) until the optical density at 600 nm ($OD_{600}$) reached 0.6. Subsequently, overexpression was induced by adding 0.5 mM isopropyl-β-D-thiogalactopyranoside for 20 hr at 20 °C. The cells were harvested by centrifugation at $4000 \times g$, resuspended in a buffer containing 50 mM Tris–HCl pH 7.4, 300 mM NaCl, DNase, and lysozyme, and disrupted by sonication. The insoluble part was removed by centrifugation at $40,000 \times g$ for 30 min at 4°C. We loaded the supernatant onto a 5-ml HisTrap HP column (GE, America). The recomEHEP was eluted using increasing concentrations of imidazole (0–500 mM). The purified proteins were dialyzed against a solution containing 50 mM Tris–HCl pH 7.4 and 50 mM NaCl and subsequently loaded onto a HiTrap Q HP column (GE, America) and eluted by a linear gradient of a solution containing 50 mM Tris–HCl and 1 M NaCl. Fractions containing recomEHEP were concentrated and then purified using a gel filtration column (HiLoad 16/60 Superdex 75 pg) (GE, America) equilibrated with 20 mM sodium acetate pH 6.0 and 100 mM NaCl. We collected the fractions containing recomEHEP and concentrated them to 2.1 mg/ml using Amicon (Merck, America).

DNA encoding *aku*BGL without a signal peptide was codon optimized (GENEWIZ, China) for overexpression in *E. coli*. The cDNA encoding the GH1D1 domain was subcloned into a modified pET-32a vector (Invitrogen, America) with an N-terminal TrxA tag and a 6×His tag, followed by a TEV protease recognition site. The primers used were shown in *Supplementary file 1*. The plasmid pET-32a-GH1D1 was electorally transformed into *E. coli* origami2 cells. We grew the cells in LB medium supplemented with 100 µg/ml of ampicillin at 37°C until the $OD_{600}$ reached 0.6. Then, the cultures were cooled and induced with 0.5 mM isopropyl-β-D-thiogalactopyranoside at 20°C for 16 hr. After resuspending the harvested cells in a buffer containing 50 mM Tris–HCl (pH 7.4), 250 mM NaCl, and 5% glycerol, we disrupted the cells by sonication. The cell lysate was centrifugated at $40,000 \times g$ for 30 min at 4°C. The supernatant was filtered by 0.45 µm membrane and then loaded onto a HisTrap HP column (GE, America). After washing with lysis buffer supplemented with 20 mM imidazole, the recomGH1D1 was eluted linearly from the column with lysis buffer supplemented with 500 mM imidazole. The fractions containing recomGH1D1 were added with TEV protease at a ratio of 1:10 and dialyzed against a buffer containing 50 mM Tris (pH 7.4), 50 mM NaCl, and 2 mM DTT at 4°C overnight. Then, we purified the recomGH1D1 by a HisTrap HP column again (GE, America) and collected the flowthrough. Finally, we purified the recomGH1D1 by size-exclusion chromatography on a HiLoad 16/60 Superdex 200 pg column (GE, America) equilibrated with a buffer containing 20 mM Bis-Tris (pH 6.0). The recomGH1D1 was concentrated to 1.5 mg/ml by centrifugation using Amicon-Ultra (molecular weight cut-off of 10 kDa).

## N-terminal sequencing of *aku*BGL

We performed an N-terminal sequencing of purified *aku*BGL using the Edman degradation method. *aku*BGL was separated by sodium dodecyl sulfate–polyacrylamide gel electrophoresis (SDS–PAGE), followed by electrophoretic transfer onto a PVDF (polyvinylidene fluoride) membrane (GE, America). The membrane was subsequently stained with Ponceau S solution. The band corresponding to *aku*BGL was excised and analyzed using a PPSQ-53A Protein sequencer (Shimadzu, Japan) at the Instrumental Analysis Service of Hokkaido University.

## Effects of TNA on *aku*BGL activity with or without EHEP

Due to the yellow color of TNA, which affects absorbance at 420 nm of the reaction product o-nitrophenol of *aku*BGL, we used high-performance liquid chromatography (HPLC) to measure the *aku*BGL

activity in the reaction system containing TNA. Ortho-nitrophenyl-β-galactoside (ONPG) was used as a substrate to measure akuBGL activity. The reaction system (100 µl) included 2.5 mM ONPG, 49 nM akuBGL, and different TNA concentrations (0, 20, 40, and 60 µM) in a reaction buffer (50 mM CH$_3$COONa pH 5.5, 100 mM NaCl, and 10 mM CaCl$_2$). After incubation for 10 min at 37°C, 100 µl of methanol was added to each sample to terminate the reaction. Then, the mixture was centrifuged for 10 min at 15,000 × g at 4°C and the supernatant was used for analyzing akuBGL activity via HPLC. To measure the protective effect of EHEP on akuBGL, we added different amounts of EHEP (1.68, 3.36, and 5.04 µM) to the reaction system (2.5 mM ONPG, 49 nM akuBGL, 40 µM TNA, 50 mM CH$_3$COONa pH 5.5, 100 mM NaCl, and 10 mM CaCl$_2$).

### RecomGH1D1 activity assay

We measured simply the activity of the recomGH1D1 using a spectrophotometer because the reaction product, o-nitrophenol, is yellow. The reaction system (100 µl) included 2.5 mM ONPG, 49 nM akuBGL or recomGH1D1 in a reaction buffer (50 mM CH$_3$COONa pH 5.5, 100 mM NaCl, and 10 mM CaCl$_2$). After incubation for 10 min at 37°C, the reaction was terminated by adding 100 µl of 500 mM Na$_2$CO$_3$. The absorbance at 420 nm was measured using a SpectraMax spectrophotometer (Molecular Devices, Japan).

### Binding assay for recomEHEP with TNA

We measured the binding activity of recomEHEP using precipitation analysis in the same method as natural EHEP, as previously described (*Tsuji et al., 2017*). Briefly, recomEHEP or EHEP was incubated with TNA at 25°C for 90 min and centrifuged for 10 min at 12,000 × g at 4°C. Then, we washed the precipitates twice and resuspended them in an SDS–PAGE loading buffer for binding analysis.

### Resolubilization of the EHEP–eckol precipitate

A mixture of 2 mg of EHEP and 0.4 mg of eckol was incubated at 37°C for 1 hr, followed by centrifugation at 12,000 × g for 10 min, and the supernatant was removed. The sediment was dissolved in a 50 mM Tris–HCl buffer at different pH (7.0–9.0), and the absorbance at 560 nm was measured over time. After checking the elution peak by SDS–PAGE, the resolubilized EHEP was analyzed by a Sephacryl S-100 HR column (2.0 × 110 cm). Moreover, the eckol-/dieckol-binding activity of resolubilized EHEP was assessed, as mentioned above in this section.

### Crystallization and data collection

The crystallization, data collection, and initial phase determination of EHEP were described previously (*Sun et al., 2020*). As EHEP precipitates when bound to TNA, we could not cocrystallize EHEP with TNA. Therefore, we used the soaking method to obtain the EHEP–TNA complex. Owing to the poor reproducibility of EHEP crystallization, we used a co-cage-1 nuclean (*Yao and Li, 2020*), a metal–organic framework (*Matsuzaki et al., 2014*) to prepare EHEP crystals for forming the complex with TNA. Finally, we obtained high-quality EHEP crystals under the reservoir solution containing 1.0 M sodium acetate and 0.1 M imidazole (pH 6.5) with co-cage-1 nuclean (*Yao and Li, 2020*). Subsequently, we soaked the EHEP crystals in a reservoir solution containing 10 mM TNA at 37°C for 2 days; then, they were maintained at 20°C for 2 weeks. Next, we soaked the EHEP crystals in a reservoir solution containing 10 mM phloroglucinol. For data collection, the crystal was soaked in a cryoprotectant solution containing 20% (vol/vol) glycerol along with the reservoir solution. Diffraction data were collected under a cold nitrogen gas stream at 100 K using Photon Factory BL-17 (Tsukuba, Japan) or Spring 8 BL-41XU (Hyogo, Japan).

For akuBGL crystallization, the initial crystallization screening was performed using the sitting-drop vapor-diffusion method with Screen Classics and Classics II crystallization kits (QIAGEN, Hilden, Germany) and PACT kits (Molecular Dimensions, Anatrace, Inc) at 20°C. Crystallization drops were set up by mixing 0.5 µl of the protein solution with an equal volume of the reservoir solution. The initial crystals were obtained under condition no. 41 (0.1 M sodium acetate pH 4.5 and 25% polyethylene glycol [PEG] 3350) of Classics II, no. 13 (0.1 M MIB buffer [25 mM sodium malonate dibasic monohydrate, 37.5 mM imidazole, and 37.5 mM boric acid] with pH 4.0 and 25% PEG 1500), and no. 37 (0.1 M MMT buffer [20 mM DL-malic acid, 40 mM MES monohydrate, and 40 mM Tris] with pH 4.0 and 25% PEG 1500) of PACT. After optimization by varying the buffer pH and precipitant concentration and

adding co-cage-1 nucleant (*Yao and Li, 2020*), the optimal crystals were obtained using 0.1 M sodium acetate pH 4.5 and 20% PEG 3350 as a reservoir solution at a protein concentration of 5.4 mg/ml with a co-cage-1 nucleant (*Yao and Li, 2020*). Diffraction data were collected under a cold nitrogen gas stream at 100 K using Photon Factory BL-1A (Tsukuba, Japan) after cryoprotection by adding glycerol to a 20% final concentration into the reservoir solution. The optimal resolution of diffraction data was obtained by soaking a crystal with 5 mM TNA in the reservoir buffer at 37°C for 4 hr.

All datasets were indexed, integrated, scaled, and merged using *XDS/XSCALE* program (*Kabsch, 2010*). Statistical data collection and process are summarized in *Table 1*.

## Structure determination and refinement

For EHEP structure determination, after initial phasing via the native-SAD method (*Sun et al., 2020*; *Yu et al., 2020*), the model was obtained and refined with *auto-building* using *Phenix AutoBuild* of the *PHENIX* software suite (*Adams et al., 2010*). The obtained native-SAD structure was used as a model for rigid body refinement using *phenix.refine* (*Afonine et al., 2012*) of the *PHENIX* software suite with the native data at a high resolution of 1.15 Å. The structure of EHEP was automatically rebuilt using *Phenix AutoBuild* of the *PHENIX* software suite again (*Adams et al., 2010*). Several rounds of refinement were performed using *phenix.refine* of the *PHENIX* software suite (*Adams et al., 2010*), alternating with manual fitting and rebuilding using *COOT* program (*Emsley and Cowtan, 2004*). The final refinement statistics and geometry are shown in *Table 1*.

The structure of the EHEP–TNA complex was determined by the molecular replacement (MR) method using the EHEP structure as a search model with *Phaser* of the *PHENIX* software suite (*McCoy et al., 2007*). The electron density block of TNA was clearly shown in both $2F_o–F_c$ and $F_o–F_c$ maps. Subsequently, the TNA structure was manually constructed, followed by several rounds of refinement using *phenix.refine* (*Adams et al., 2010*), with manual fitting and rebuilding using *COOT* (*Emsley and Cowtan, 2004*). We also determined the structure of phloroglucinol-soaked crystals at a resolution of 1.4 Å by the MR method using the refined EHEP structure as a search model with *phenix.phaser*. However, no electron density block of phloroglucinol was obtained. Therefore, we referred to this structure as the apo form (apo structure2). The final refinement statistics and geometry are shown in *Table 1*.

We determined the structure of *aku*BGL by the MR method using *Phaser* of the *PHENIX* software suite (*McCoy et al., 2007*). We used one GH domain (86–505 aa) of β-klotho (PDB entry: 5VAN) (*Lee et al., 2018*) as the search model. The GH domain of β-klotho shares 30% sequence identity with *aku*BGL. Four GH domains of two molecules were found in an asymmetric unit and rebuilt with *Phenix_autobuild* of *Phenix* software suite (*Adams et al., 2010*). Finally, refinement of *aku*BGL structure was performed as described for EHEP.

## Docking studies of *aku*BGL with phlorotanins and laminarins

We used the Schrodinger Maestro program to perform docking studies (*Sastry et al., 2013*). First, we superimposed the structure of the *Os*BGL mutant complexed with cellotetraose (PDB ID 4QLK; *Pengthaisong and Ketudat Cairns, 2014*) onto that of *aku*BGL GH1D2 to define the ligand position in the ligand-binding cavity. Then, we modified the structure of the *aku*BGL GH1D2 using the wizard module to remove water molecules and add hydrogen atoms for docking. The 2D structures of the inhibitory ligands, including TNA, phloroglucinol, and eckol, were downloaded from PubChem (*Wang et al., 2009*) and further converted to 3D structures using the LigPrep module of the Schrodinger Maestro program. The structure of the substrate laminaritetraose was extracted from the *Zobellia galactanivorans* β-glucanase–laminaritetraose complex structure (PDB ID: 4BOW; *Labourel et al., 2014*). Then, a receptor grid was constructed in the center of the ligand-binding cavity. We performed docking using the Glide standard precision mode without any constraints. The optimal binding pose was determined using the lowest Glide score, and the docked structures were analyzed using PyMol.

## Acknowledgements

This work was supported in part by Grant-in-Aid for Scientific Research (B) (Grant Number 21H01754 to MY) and Platform Project for Supporting Drug Discovery and Life Science Research (Basis for Supporting Innovative Drug Discovery and Life Science Research (BINDS)) from Japan Agency for Medical Research and Development (AMED) under Grant Number JP18am0101071 and

JP19am0101083. We are grateful to the Photon Factor and SPring-8 (No. 2017B2545, 2017A2551, and 2018B2538) for beam time and the beamline staff for their assistance with data collection.

## Additional information

### Funding

| Funder | Grant reference number | Author |
|---|---|---|
| Japan Society for the Promotion of Science | 21H01754 | Min Yao |
| Japan Agency for Medical Research and Development | JP18am0101071 | Min Yao |
| Japan Agency for Medical Research and Development | JP19am0101083 | Min Yao |

The funders had no role in study design, data collection, and interpretation, or the decision to submit the work for publication.

### Author contributions

Xiaomei Sun, Validation, Investigation, Visualization, Writing – original draft, Writing – review and editing; Yuxin Ye, Koji Kato, Jian Yu, Investigation, Visualization, Writing – review and editing; Naofumi Sakurai, Investigation; Hang Wang, Keizo Yuasa, Investigation, Writing – review and editing; Akihiko Tsuji, Conceptualization, Validation, Investigation; Min Yao, Conceptualization, Data curation, Supervision, Funding acquisition, Validation, Investigation, Visualization, Writing – original draft, Project administration, Writing – review and editing

### Author ORCIDs

Xiaomei Sun ⃝ http://orcid.org/0000-0001-5228-5333
Hang Wang ⃝ http://orcid.org/0000-0001-7706-0536
Keizo Yuasa ⃝ https://orcid.org/0000-0003-4831-0465
Min Yao ⃝ http://orcid.org/0000-0003-1687-5904

Reviewer #1 (Public Review): https://doi.org/10.7554/eLife.88939.3.sa1
Reviewer #2 (Public Review): https://doi.org/10.7554/eLife.88939.3.sa2
Reviewer #3 (Public Review): https://doi.org/10.7554/eLife.88939.3.sa3
Author Response https://doi.org/10.7554/eLife.88939.3.sa4

## Additional files

### Supplementary files
• Supplementary file 1. Primers used in this study.
• MDAR checklist

### Data availability

The atomic coordinates were deposited in the PDB with the accession codes as follows: EHEP with 1.15 Å resolution (8IN3), EHEP with 1.4 Å resolution (8IN4), EHEP complexed with tannic acid (8IN6), akuBGL (8IN1).

The following datasets were generated:

| Author(s) | Year | Dataset title | Dataset URL | Database and Identifier |
|---|---|---|---|---|
| Sun XM, Ye YX, Kato K, Yu J, Yao M | 2023 | Eisenia hydrolysis-enhancing protein from Aplysia kurodai | https://www.rcsb.org/structure/8IN3 | RCSB Protein Data Bank, 8IN3 |
| Sun XM, Ye YX, Kato K, Yu J, Yao M | 2023 | Eisenia hydrolysis-enhancing protein from Aplysia kurodai | https://www.rcsb.org/structure/8IN4 | RCSB Protein Data Bank, 8IN4 |
| Sun XM, Ye YX, Kato K, Yu J, Yao M | 2023 | Eisenia hydrolysis-enhancing protein from Aplysia kurodai with tannic acid | https://www.rcsb.org/structure/8IN6 | RCSB Protein Data Bank, 8IN6 |
| Sun XM, Ye YX, Kato K, Yu J, Yao M | 2023 | beta-glucosidase protein from Aplysia kurodai | https://www.rcsb.org/structure/8IN1 | RCSB Protein Data Bank, 8IN1 |

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
