## [Editor Report · eLife assessment]

This **important** study presents **convincing** evidence on how the sea slug *Aplysia kurodai* optimizes its digestion of brown algae, in a classical predator-prey 'arms race' at the molecular level. The experimental protein structures and enzyme assays provide support for the claims of how *A. kurodai* avoids inhibition by algal compounds, and also hold promise for biotechnological applications.

---

## [Referee Report · Reviewer #1 (Public Review)]

Sun and co-authors have determined the crystal structures of EHEP with/without phlorotannin analog, TNA, and akuBGL. Using the akuBGL apo structure, they also constructed model structures of akuBGL with phlorotannins (inhibitor) and laminarins (substrate) by docking calculation. They clearly showed the effects of TNA on akuBGL activity with/without EHEP and resolubilization of the EHEP-phlorotannin (eckol) precipitate under alkaline conditions (pH >8). Based on this knowledge, they propose the molecular mechanism of the akuBGL-phlorotannin/laminarin-EHEP system at the atomic level. Their proposed mechanism is useful for further understanding of the defensive-offensive association between algae and herbivores.

---

## [Referee Report · Reviewer #2 (Public Review)]

In this study the authors try to understand the interaction of a 110 kDa ß-glucosidase from the mollusk Aplysia kurodai, named akuBGL, with its substrate, laminarin, the main storage polysaccharide in brown algae. On the other hand, brown algae produce phlorotannin, a secondary metabolite that inhibits akuBGL. The authors study the interaction of phlorotannin with the protein EHEP, which protects akuBGL from phlorotannin by sequestering it in an insoluble complex.

The strongest aspect of this study is the outstanding crystallographic structures they obtained, including akuBGL (TNA soaked crystal) structure at 2.7 Å resolution, EHEP structure at 1.15 Å resolution, EHEP-TNA complex at 1.9 Å resolution, and phloroglucinol soaked EHEP structure at 1.4 Å resolution. EHEP structure is a new protein fold, constituting the major contribution of the study.

The drawback on EHEP structure is that protein purification, crystallization, phasing and initial model building were published somewhere else by the authors, so this structure represents incremental research.

One concern remains unanswered to me. If the mechanism of action of EHEP is to precipitate together with TNA in a 1:1 insoluble complex, then it does not matter if there are multiple mechanisms involved in the activity assay, the protection of 4uM EHEP against 40uM TNA simply requires a different stoichiometry.

---

## [Referee Report · Reviewer #3 (Public Review)]

The manuscript by Sun et al. reveals several crystal structures that help underpin the offensive-defensive relationship between the sea slug Aplysia kurodai and algae. These centre on TNA (a algal glycosyl hydrolase inhibitor), EHEP (a slug protein that protects against TNA and like compounds) and BGL (a glycosyl hydrolase that helps digest algae). The hypotheses generated from the crystal structures herein are supported by biochemical assays.

The crystal structures of apo and TNA-bound EHEP reveals the binding (and thus protection) mechanism. The authors then demonstrate that the precipitated EHEP-TNA complex can be resolubilised at an alkaline pH, potentially highlighting a mechanism for EHEP recycling in the A. kurodai midgut. The authors also present the crystal structures of akuBGL, a beta-glucosidase utilised by Aplysia kurodai to digest laminarin in algae into glucose. The structure revealed that akuBGL is composed of two GH1 domains, with only one GH1 domain having the necessary residue arrangement for catalytic activity, which was confirmed via hydrolytic activity assays. Docking was used to assess binding of the substrate laminaritetraose and the inhibitors TNA, eckol and phloroglucinol to akuBGL. The docking studies revealed that the inhibitors bound akuBGL at the glycone-binding suggesting a competitive inhibition mechanism. Overall, most of the claims made in this work are supported by the data presented.

---

## [Author Response]

The following is the authors’ response to the original reviews.

**Reviewer #1 (Public Review):**
Sun and co-authors have determined the crystal structures of EHEP with/without phlorotannin analog, TNA, and akuBGL. Using the akuBGL apo structure, they also constructed model structures of akuBGL with phlorotannins (inhibitor) and laminarins (substrate) by docking calculation. They clearly showed the effects of TNA on akuBGL activity with/without EHEP and resolubilization of the EHEP-phlorotannin (eckol) precipitate under alkaline conditions (pH >8). Based on this knowledge, they propose the molecular mechanism of the akuBGL-phlorotannin/laminarin-EHEP system at the atomic level. Their proposed mechanism is useful for further understanding of the defensive-offensive association between algae and herbivores. However, there are several concerns, especially about structural information, that authors should address.

Thank you for reviewing our manuscript. We addressed all comments below.

1. TNA binding to EHEPThe electron densities could not show the exact conformations of the five gallic acids of TNA, as the authors mentioned in the manuscript. On the other hand, the authors describe and discuss the detailed interaction between EHEP and TNA based on structural information. The above seems contradictory. In addition, the orientation of TNA, especially the core part, in Fig. 4 and PDB (8IN6) coordinates seem inconsistent. The authors should redraw Fig. 4 and revise the description accordingly to be slightly more qualitative.

We apologize for the mistake with the PDB file. We forgot to re-upload the final coordinate file of 8IN6, which had been modified according to the requirement of the PDB instructions. We have now re-uploaded the correct PDB file. We carefully checked Fig. 4 (Fig.3 in the revised version), which used the final coordinate file of 8IN6.

1. Two domains of akuBGLThe authors concluded that only the GH1D2 domain affects its catalytic activity from a detailed structural comparison and the activity of recombinant GH1D1. That conclusion is probably reasonable. However, the recombinant GH1D2 (or GH1D1+GH1D2) and inactive mutants are essential to reliably substantiate conclusions. The authors failed to overexpress recombinant GH1D2 using the *E. coli* expression system. Have the authors tried GH1D1+GH1D2 expression and/or other expression systems?

By referencing other BGLs (six samples were expressed by using *E. coli*, and one was expressed by using Pichia), we only tried the overexpression of akuBGL, GH1D1, GH1D2, and GH1D1+GH1D2 in *E. coli* expression system using several different vectors. As the reviewer mentioned that inactive mutants are essential to substantiate our conclusion reliably, it will be tried further to use yeast or cell expression systems to confirm our conclusion. We added these limitations as “Future assay of GH1D2 and inactive mutants is the complement to validate the molecular mechanism of akuBGL” in the discussion (Line 343-345)

1. Inhibitor binding of akuBGLThe authors constructed the docking structure of GH1D2 with TNA, phloroglucinol, and eckol because they could not determine complex structures by crystallography. The molecular weight of akuBGL would also allow structure determination by cryo-EM, but have the authors tried it? In addition, the authors describe and discuss the detailed interaction between GH1D2 and TNA/phloroglucinol/eckol based on docking structures. The authors should describe the accuracy of the docking structures in more detail, or in more qualitative terms if difficult.

Yes, it is possible to try cryo-EM for obtaining the structure of akuBGL complexed with the ligand. However, we didn’t try because 110 kDa akuBGL consists of two 55 kDa GH1Ds linked by along loop, and we worried that ligand may not be visualized using cryo-EM.

Following the comment, we added the description of the accuracy of the docking structures as “Those docking scores corroborated well with the inhibition activity toward akuBGL, that TNA had a more robust inhibition activity than phloroglucinol, indicating that the docking results are reasonable.” (Line 322-324)

**Reviewer #2 (Public Review):**
In this study the authors try to understand the interaction of a 110 kDa ß-glucosidase from the mollusk Aplysia kurodai, named akuBGL, with its substrate, laminarin, the main storage polysaccharide in brown algae. On the other hand, brown algae produce phlorotannin, a secondary metabolite that inhibits akuBGL. The authors study the interaction of phlorotannin with the protein EHEP, which protects akuBGL from phlorotannin by sequestering it in an insoluble complex.The strongest aspect of this study is the outstanding crystallographic structures they obtained, including akuBGL (TNA soaked crystal) structure at 2.7 Å resolution, EHEP structure at 1.15 Å resolution, EHEP-TNA complex at 1.9 Å resolution, and phloroglucinol soaked EHEP structure at 1.4 Å resolution. EHEP structure is a new protein fold, constituting the major contribution of the study.

We thank you for reviewing our manuscript.

The drawback on EHEP structure is that protein purification, crystallization, phasing and initial model building were published somewhere else by the authors, so this structure is incremental research and not new.

We have published the results of protein purification, crystallization, phasing, and initial model building for determining structure but have yet to give the structure since further structural refinement is indispensable. Such published data in [Acta F] is a service for obtaining the structure.

We believe that the structure of the EHEP holds great importance, and it is the first time to publish.

Most of the conclusions are derived from the analysis of the crystallographic structures. Some of them are supported by other experimental data, but remain incomplete. The impossibility to obtain recombinant samples, implying that no mutants can be tested, makes it difficult to confirm some of the claims, especially about the substrate binding and the function of the two GH1Ds from akuBGL.

As mentioned by the reviewer, mutant analysis would be the best way to substantiate our conclusions. However, it is challenging to obtain recombinant samples, although we tried to overexpress them (akuBGL, GH1D1, GH1D2, and GH1D1+GH1D2). So, we did the structural comparison, and docking simulation to propose the molecular mechanism. We added these limitations as “Further assay of GH1D2 and inactive mutants is the complement to validate the molecular mechanism of akuBGL” in the discussion part (Line 343-345).

The authors hypothesize from their structure that the interaction of EHEP with phlorotannins might be pH dependent. Then they succeed to confirm their hypothesis, showing they can recover EHEP from precipitates at alkaline pH, and that the recovered EHEP can be reutilized.A weakness in the model is raised by the fact that the stoichiometry of the complex EHEP:TNA is proposed to be 1:1, but in Figure 1 they show that 4 µM of EHEP protects akuBGL from 40 µM TNA, meaning EHEP sequesters more TNA than expected, this should be addressed in the manuscript.

The assay experiment in figure1 does not directly provide the stoichiometric ratio of EHEP: TNA because the activity assay system consists of substrate of akuBGL, akuBGL, TNA, and EHEP, which involves multiple equilibration processes: akuBGL⇋ substrate, akuBGL⇋TNA, and EHEP ⇋TNA. To avoid misunderstanding, we added the descriptions of ″As this activity assay system involves multiple equilibration processes: akuBGL⇋substrate, akuBGL⇋TNA, and EHEP ⇋TNA.″(Line 120-121).

The authors study the interaction of akuBGL with different ligands using docking. This technique is good for understanding the possible interaction between the two molecules but should not be used as evidence of binding affinity. This implies that the claims about the different binding affinities between laminarin and the inhibitors should be taken out of the preprint.

Following the suggestion, we deleted the descriptions about the difference in binding affinity with docking scores at the last paragraph of [Inhibitor binding of akuBGL].

In the discussion section there is a mistake in the text that contradicts the results. It is written "EHEP-TNA could not dissolve in the buffer of pH > 8.0" but the result obtained is the opposite, the precipitate dissolved at alkaline pH.

We apologize for this mistake and corrected it to " EHEP–TNA could dissolve in the buffer of pH > 8.0." (Line 394).

Solving a new protein fold, as the authors report for EHEP, is relevant to the community because it contributes to the understanding of protein folding. The study is also relevant dew to the potential biotechnological application of the system in biofuel production. The understanding on how an enzyme as akuBGL can discriminate between substrates is important for the manipulation of such enzyme in terms of improving its activity or changing its specificity. The authors also provide with preliminary data that can be used by others to produce the proteins described or to design a strategy to recover EHEP from precipitates with phlorotannin at industrial scales.In general methods are not carefully described, the section should be extended to improve the manuscript.

Following the comment, we added the method descriptions

1. Recombinant GH1D1 domain expression and purification in [EHEP and akuBGL preparation].

2. Sections of [recomGH1D1 activity assay], and [N-terminal sequencing of akuBGL]

3. More details of resolubiliztion of EHEP and activity in [Resolubilization of the EHEP–eckol precipitate].

**Reviewer #3 (Public Review):**
The manuscript by Sun et al. reveals several crystal structures that help underpin the offensivedefensive relationship between the sea slug Aplysia kurodai and algae. These centre on TNA (a algal glycosyl hydrolase inhibitor), EHEP (a slug protein that protects against TNA and like compounds) and BGL (a glycosyl hydrolase that helps digest algae). The hypotheses generated from the crystal structures herein are supported by biochemical assays.The crystal structures of apo and TNA-bound EHEP reveals the binding (and thus protection) mechanism. The authors then demonstrate that the precipitated EHEP-TNA complex can be resolubilised at an alkaline pH, potentially highlighting a mechanism for EHEP recycling in the A. kurodai midgut. The authors also present the crystal structures of akuBGL, a beta-glucosidase utilised by Aplysia kurodai to digest laminarin in algae into glucose. The structure revealed that akuBGL is composed of two GH1 domains, with only one GH1 domain having the necessary residue arrangement for catalytic activity, which was confirmed via hydrolytic activity assays. Docking was used to assess binding of the substrate laminaritetraose and the inhibitors TNA, eckol and phloroglucinol to akuBGL. The docking studies revealed that the inhibitors bound akuBGL at the glycone-binding suggesting a competitive inhibition mechanism. Overall, most of the claims made in this work are supported by the data presented.

We thank you very much for reviewing our manuscript.

**Reviewer #1 (Recommendations For The Authors):**
• Fig. 3 should be moved to the Supplements because acetylation modification at the N-terminus is not essential for the function of EHEP.

Following the recommendation, we moved Fig.3 to Supplements (Fig. S2).

• EHEP2 is processed at 1.4 Å resolution, however, the statistics at highest resolution shell indicate you can process at higher resolution. Why 1.4 Å resolution?

We tried to process this dataset at the higher resolution at 1.35 Å, and the completeness and I/sigma of the highest resolution shell reduced to 88.9% and 2.16, respectively. The parameter of I/sigma is OK, but the completeness reduced seriously. So, we set a cutoff of 1.4 Å.

• Fig. S1A should be revised to include the gallic acid numbers (1, 2, 3, 4, 6) and the 3.0 σ map. >

As presented in Fig. S1A, the omitted map (fo–fc map) of the ligand TNA, countered at 2.0 σ, showed that gallic acid 2 has poor density, and gallic acid 4 has weak density. Moreover, the TNA is relatively big to EHEP (7.5 %), and the omitted map countered 3.0 σ could not clearly show gallic acids. So, we keep the map at 2.0 σ in Fig. S3A.

• The authors should provide more information on "co-cage-1 nucleant".

Our lab is currently publishing a paper that provides detailed information on the co-cage-1 nucleant, including components, synthesis, nucleation mechanism, and application. Once the paper is published, we will cite it in this manuscript.

**Reviewer #2 (Recommendations For The Authors):**
Is the word "offence" the appropriate word for referring to the activity of EHEP? Is this word used in the literature for this system? I find it confusing but might be because I am not in the specific topic.

In the field of prey–predator, the defense–offensive is commonly used.

According to Charles D. Amsler's book ″Algal Chemical ecology″, Herbivore offensive is the traits that allow herbivores to increase feeding rates on algae. Therefore, in our opinion, the offensive is appropriate.

Taking into consideration that I am not an English language expert I find the writing of the manuscript could be improved in general. Here are some lines as examples of where the grammar could be better:Line 193: "decrement of the loop part"

Following the comment, we corrected it to "decrease of the loop part" (Line 197).

Line 199: there is a typographical error.

We apologize for our mistake and corrected it to “EHEP” (Line 202).

Line 205-206: "only hydrophobically interacted with"

Following the comment, we modified it to "only interacted hydrophobically with EHEP" (Line 209)

Line 224: "phlorotannin–precipitate activity"

Following the comment, we modified it to “phlorotannin-precipitate activity” (Line 227).

Line 232: "without the N-terminal 25 residues"

Following the comment, we modified it to "lacked the N-terminal 25 residues" (Line 236).

Line 353: "bound" should be "bind"

We apologize for our mistake and modified it (Line 356).

Line 359: "predator mammals"

We apologize for our mistake and modified it to "predatory mammals" (Line 363).

Line 363: "at an alkaline pH of insect midgut"

Following the comment, we modified it to "at the alkaline pH of the insect midgut" (Line 367).

Line 370: "nonstructural proteins" means "unstructured proteins"?

Yes, unfolding proteins, we modified to "unfolding proteins with randomly coils" (Line 374).

Line 374: "similar strategy with mammals"

Following the comment, we modified it to "similar strategy to mammals" (Line 379).

Line 403: "to forming"

We apologize for our mistake and modified it to "to form" (Line 404).

Line 404: "considered no binding"

We apologize for our mistake and modified it to "considered not binding" (Line 405).

Line 406: "activity pocket" means the active site?

Yes, we modified it to "active site" (Line 407).

Line 424: "step purification"

Following the comment, we corrected it to "one step for purification" (Line 425).

Line 431

Following the comment, we corrected it to “To verify whether the chemical modifications which was indicated by previous study affects” (Line 432-433).

Line 812: there is typographical error

We apologize for our mistakes, and corrected it to Tris-HCl” for all “Tris–HCl (Line 878~).

Line 223: eckol is not mentioned in the text and appears for the first time in the figure caption.

Following the comment, we added “eckol” in the first section of the [Result] (Line 117).

The paragraph between lines 271 and 280 is disconnected from the previous one and it is not about results, it should be at the discussion section.

Following the comment, we moved them to the discussion part (Line 335-343).

Line 324: "the three inhibitors inhibited": this claim should be corrected to "the three inhibitors interacted", since the word inhibited would imply the authors measured activity experimentally.

We modified it as the comment. (Line 325).

Line 392: "could not dissolve" is contradicting the result.

We apologize for our mistake and corrected it to "could dissolve" (Line 394).

They describe acetylation but they try overexpressing in *E. coli*, could it be that they needed to express the construct in a system where they would get the acetylation? At least this should be discussed in the text.

Because our sample of EHEP with acetylation was purified from the natural source of the digestive fluid of A.kurodai, we only need to express EHEP without acetylation. Following the comment, we modified the descriptions to clarify it in the section (Lines 170-173 and 177-179).

“Consistent with the molecular weight results obtained using MALDI–TOF MS, the apo structure2 (1.4 Å resolution) clearly showed that the cleaved N-terminus of Ala21 underwent acetylation, demonstrating that EHEP is acetylated in A. kurodai digestive fluid.”

"To explore whether acetylation affects the protective effects of EHEP on akuBGL, we used the *E. coli* expression system to obtain the unmodified recomEHEP (A21–K229)."

From the text it is not clear in which biological context the brown algae meet the attack by the hydrolase, the information is spread all over the manuscript, it should be clearly described at the introduction.

When the brown algae are consumed as food by sea hare A. kurodai, they meet the attack by the hydrolase akuBGL. Following the comment, we clear the descriptions in the introduction part as below (Line 42-45).

″In brown algae Eisenia bicyclis, laminarin is a major storage carbohydrate, constituting 20%–30% of algae dry weight. The sea hare Aplysia kurodai, a marine gastropod, preferentially feeds on the E. bicyclis with its 110 and 210 kDa β-glucosidases (akuBGLs), hydrolyzing the laminarin and releasing large amounts of glucose.″

Affinity ranking based on docking is not reliable, the differences in free energy are in the same order of magnitude. I would recommend erasing this claim since it is not fundamental to the study.Another option would be to determine affinities experimentally.

We agree with the comment and removed the text about affinity ranking with docking scores.

Figure 1: relative activity is not defined. HPLC data should be shown as supplementary material.

Following the comment, we added the definition of relative activity and the HPLC data as Fig. S1 in the revised version.

Figure 4: Sephacryl resin is mentioned here but not described in the methods.

Following the comment, we added the description in the methods (Line 515).

Protein N-terminal sequencing analysis should be described in the methods.

Following the comment, we added the sequencing analysis in the methods (Line 476-483).

Figure S1 C: it should be specified how the surface electrostatic potential at different pH was calculated.

Following the comment, we added the descriptions of how the surface electrostatic potential at different pH was calculated in the figure legend of Fig. S2 of the revised version (Line 876-877).

Since the authors are capable of producing good amounts of akuBGL and have already conducted glycosidase activity assays using ONPG, it would not be difficult for them to run some kinetics experiments for the enzyme in the presence of the different inhibitors to confirm their hypothesis derived from the docking calculations.

As mentioned by the reviewer, kinetics experiments are the best way to confirm our hypothesis derived from docking calculations. However, the yield of akuBGL purification from the digestive fluid of sea hare A.kurodai is quite difficult. We could not obtain a sufficient sample of akuBGL to conduct the kinetic experiments. So, we stopped at docking simulation in this study. We added such limitations of ″Future kinetic experiments are required to validate quantitatively the competitive inhibition of phlorotannin against akuBGL″ (Line 359-360).

Some citations are missing in the discussion section, for example in lines 362, 364 and 396.

Following the comment, we added the citations.

**Reviewer #3 (Recommendations For The Authors):**
Please see comments/suggestions below for revisions.Line 176-178 - Text explains that recombEHEP precipitated after incubation with TNA to a comparable level to natural EHEP. However, figure 3B shows no comparison between recombinant and natural EHEP.

As the reviewer suggested, we repeated the binding assay of recomEHEP to confirm the precipitation with TNA and added a precipitation result of natural EHEP (Fig. S2B right) for comparing.

Line 223 - The work presented in Figure S1E goes partway towards demonstrating the activity of resolubilised EHEP. This claim would be strengthened if resolubilised EHEP was used in the akuBGL Galactoside hydrolytic activity assay and is then seen to rescue akuBGL activity in the presence of TNA.

Yes, our claim would be strengthened by adding resolubilized EHEP to akuBGL assay in the presence of TNA. Since we have obtained and presented the relationship between the precipitating of EHEP with TNA and the rescuing akuBGL activity from TNA, we only used the precipitation to demonstrate the activity of resolubilized EHEP.

Line 380-384 - Here it is discussed how TNA simultaneously binds to three EHEP molecules thus crosslinking them. It is then proposed that this could be the mechanism of precipitation. However, it is noted that TNA is soaked into crystals, therefore it is likely that this lattice exists whether TNA is present or not (this absolutely needs to be mentioned in the text). It would be possible to test this mechanism through mutagenesis. If the sites where TNA packs in between chains of EHEP were mutated to prevent crosslinking, it could then be determined whether crosslink-null EHEP can still precipitate TNA.

As the review mentioned, we do not have enough experiments to propose that the TNA-crosslink may cause the EHEP-TNA precipitation. So, we deleted the discussion of the TNA crosslink and the corresponding figure.

All docked models need to be deposited (perhaps modelarchive.org) and this resource referred to in the text.

The structures in modelarchive.org site are either homology models or de novo. We think the docked model is out of this site. So, we did not deposit them.

The x-ray data table contains data previously published in the referenced Acta cryst publication.What is eLife policy on this "double use" of data?

We apologize for our mistake, and deleted the SAD data in Table 1.

Minor pointsLine 26 - use "apo akuBGL" so as not to infer a tannic-acid bound form of this also >

Following the comment, we modified it to “apo akuBGL” (Line 26).

Line 48 - The sentence currently reads as A. kurodai is being digested.

Following the comment, we modified it to “by A. kurodai” (Line 48).

Line 49-50 & Line 65-66 - Both these lines make the same point about the impact of phlorotannin inhibition on the use of brown algae as feedstocks for biofuel, please remove one.

Following the comment, we deleted the line 49-50.

Line 115 - This needs attention as its an unusual opening sentence

Following the comment, we modified it o “Phlorotannin, a type of tannin, is a chemical defense metabolite of brown algae.” (Line 114).

Line 130 - Should the EHEP concentration be 3.96 µM not 3.36?

We apologize for our mistake 3.36 is correct, and we corrected the X-axis label in Fig.1B.

Line 133 - consider using "non-recombinant" rather than "natural"

To distinguish between non-recombinant and recombinant samples, we used “EHEP” and “akuBGL” as purified from the native source and recomEHEP and recomakuBGL as the samples overexpressed from *E. coli* in this manuscript. So, we added the definition in [Introduction] (Line 100-101).

Line 134 - "The residues A21-V227 of A21-K229..." This sentence could be written more clearly.

Following the comment, we re-wrote it to “The residues A21–V227 in purified EHEP (1–20 aa were cleaved during maturation) were built” (Line 135-136).

Line 136 - switch "appropriately visualized" for "tracable"?

Following the comment, we modified it to “built” (Line 136).

Line 158 - use "70% of backbone in a loop conformation" >

We modified as the comment (Line 159-160).

Line 184 - reword "map showed an electron density blob". (Map showed positive electron density)

Following the comment, we modified it to “map showed the electron density” (Line 188).

Line 193-194 - Is EHEP really more stable when bound to TNA? It is not shown experimentally?It is difficult to see which loop changes. Is the difference a result of crystal packing? Please switch "decrement" for another term

The regions with conformation change between EHEP and EHEP–TNA are close to TNA but not at the intermolecular interface. As the reviewer mentioned, we could not clarify the EHEP stability depended on TNA-binding, and deleted the descriptions in the second paragraph of [TNA binding to EHEP].

Following the comment, we redraw Fig. S1B (Fig. S3B in the revised version) to show the conformation changes clearly. We also modified "decrement" to "decrease" (Line 197).

Fig S1B - Can an extra figure be added to show the secondary differences more clearly? >

We redraw this figure (Fig. S3B) using closeup view to show the differences.

Line 212-213 - There is a slight discrepancy between the text and Figure 4B. Gallic acid 4 interacts with P201 and gallic acid 6 interacts with P77.

We apologize for our mistake in the text. and corrected it to “gallic acid4 and 6 showed alkyl–π interaction with P201 and P77, respectively” (Line 216).

Figure 4D - Change x axis from tube number to elution volume. Both chromatograms could also be superimposed for interpretability.

Since we used raw data from the experiment, we kept the x-axis in tube number with additional “2.7 ml/tube” information (Fig.3D).

Line 229 - Please change "there was no blob of TNA in the electron density" to there was no electron density for TNA or something similar.

Following comment, we modified it to “there was no electron density of TNA or something similar in the 2Fo–Fc and Fo–Fc map” (Line 232).

Line 231 - asymmetric unit is a more standard term (also in Fig S2 legend)

We modified as the comment (Line 235 and 885).

Line 234-235 - Reword "the residues L26-P978 of L26-N994" to make it more concise. >

Following the comment, we deleted “of L26-N994” (Line 239).

Lines 296-299 could be written more carefully - pi stacking with what? >

We apologize for our mistake and corrected it to CH–π� (Line 293).

Line 349 - which putatively enables it to......

We modified it as the commend (Line 353 in the revised manuscript).

Line 370 - "nonstructural" is the wrong term because they remain structured - use something akin to non-classical secondary structure

Following the comment, we modified it to“are unfolding proteins with randomly coils in solution " (Line 374)

Throughout - use phenix autobuild, not autobuil

We apologize for our mistakes and corrected them throughout the manuscript.

Figure 1 - the graphs would be more interpretable with all data points shown overlaid

The two graphs in Figure 1 showed two experiments with different reaction conditions. Figure 1A presents various TNA concentrations, while Figure 1B maintains a constant concentration of 40 μM for TNA with varying EHEP concentrations. So, overlaying the graphs is not feasible. Therefore, we would like to keep them separated and added the reaction condition in figure legend.

Figure 4 - in part D add an extra statement outlining what the S-100 analysis demonstrated

S-100 analysis is using a gel filtration column with Sephacryl S-100 media. We added an extra statement in the method and the legend (Fig. 3, Lines 515 and 879).

Figure 5 (and elsewhere) - the structures referred to need a PDB code and reference given in legend

Following the comment, we checked the manuscript carefully and added PDB code to the referred structures.

Fig S1 - please add an additional panel showing part D but in proper structure form, not schematic shapes

Since we do not have enough experiments to validate the TNA-crosslink, we deleted the discussion of the TNA crosslink and Fig. S1D.

Figure sig 4 - Text contains in depth information of side chain hydrogen bonding and π-π interactions between akuBGL and laminarittrose. However, the figure only shows a surface model.Consider adding a figure showing these interactions.

Following the suggestion, we added a closeup view to show these detailed interactions (Fig. S6B).